



# LakeBeD-US: a benchmark dataset for lake water quality time series and vertical profiles

Bennett J. McAfee[1], Aanish Pradhan[2], Abhilash Neog[2], Sepideh Fatemi[2], Robert T. Hensley[3], Mary E. Lofton[4], Anuj Karpatne[2], Cayelan C. Carey[4], and Paul C. Hanson[1]

[1]Center for Limnology, University of Wisconsin–Madison, Madison, WI 53706, USA
[2]Department of Computer Science, Virginia Tech, Blacksburg, VA 24061, USA
[3]National Ecological Observatory Network – Battelle, Boulder, CO 80301, USA
[4]Department of Biological Sciences, Virginia Tech, Blacksburg, VA 24061, USA

**Correspondence:** Bennett J. McAfee (bennettjmcafee@gmail.com)

**Abstract.** Water quality in lakes is an emergent property of complex biotic and abiotic processes that differ across spatial and temporal scales. Water quality is also a determinant of ecosystem services that lakes provide, and thus is of great interest to ecologists. Increasingly, machine learning and other computer science techniques are being used to predict water quality dynamics as well as to gain a greater understanding of water quality patterns and controls. To benefit both the sciences of ecology and computer science, we have created a benchmark dataset of lake water quality time series and vertical profiles. LakeBeD-US contains over 500 million unique observations of lake water quality collected by multiple long-term monitoring organizations across 17 water quality variables in 21 lakes in the United States. There are two published versions of LakeBeD-US: an "Ecology Edition" published in the Environmental Data Initiative repository, and a "Computer Science Edition" published in the Hugging Face repository. Each edition is formatted in a manner conducive to inquiries and analyses specific to each domain. For ecologists, LakeBeD-US provides an opportunity to study the spatial and temporal dynamics of several lakes with varying water quality, ecosystem, and landscape characteristics. For computer scientists, LakeBeD-US acts as a benchmark dataset that enables the advancement of machine learning for water quality prediction.

## 1 Introduction

Water quality is a critical determinant of the ecosystem services provided by lakes (Keeler et al., 2012; Angradi et al., 2018). Water quality varies across spatial and temporal scales (Hanson et al., 2006; Langman et al., 2010; Soranno et al., 2017) due to a variety of interacting physical and biological processes. For example, hypolimnetic anoxia (low oxygen) in lakes decreases habitat for cold-water fish species (Arend et al., 2011; Jane et al., 2024). Anoxia can be fueled by the product of another water quality problem, the formation of toxic phytoplankton blooms (Jane et al., 2021). Both of these water quality phenomena emerge at the ecosystem scale as a consequence of multiple physical-biological interactions, driven by external nutrient loads and weather conditions (Paerl and Huisman, 2009; Snortheim et al., 2017; Ladwig et al., 2021; Jane et al., 2021). While there is mechanistic understanding of how these water quality phenomena evolve for well-studied lake systems, predicting their occurrence under scenarios of change or in large numbers of systems with sparse data remains challenging (Guo et al., 2021;





Miller et al., 2023). To meet this challenge, we need scalable water quality models that are supported by observational data of sufficient spatiotemporal resolution to recreate key water quality dynamics (Ejigu, 2024; Varadharajan et al., 2022).

Knowledge-guided machine learning (KGML) has emerged as a powerful technique for incorporating both ecological knowledge and observational data within a model (Karpatne et al., 2017, 2024). By fusing machine learning with physical and ecological principles, KGML has proven effective for assessing lake surface area change (Wander et al., 2024), modeling lake temperature (Read et al., 2019; Daw et al., 2014; Ladwig et al., 2024; Chen et al., 2024b), phytoplankton (chlorophyll) forecasting (Lin et al., 2023; Chen et al., 2024a), and predicting lake phosphorus concentrations (Hanson et al., 2020). As has been 30 shown, a variety of modeling techniques within and beyond KGML are required to advance water quality understanding and prediction (Wai et al., 2022; Lofton et al., 2023). Creative approaches will likely spring from interdisciplinary collaborations of both lake ecologists and computer scientists (Carey et al., 2019) and will need diverse, high volume, high quality observational data that are easily accessible to researchers from multiple disciplines.

Predicting the evolution of water quality through time and space requires treating lakes as dynamical systems that operate 35 across many scales. Studies that have addressed the temporal dynamics of water quality at broad spatial scales are few in number (but see, Wilkinson et al., 2022; Zhao et al., 2023; Meyer et al., 2024), due in large part to the nature of data collection and research project design focusing on one scale at a time. Datasets that capture spatial gradients (Soranno et al., 2017; Pollard et al., 2018), temporal gradients (Magnuson et al., 2006; Goodman et al., 2015), or both have been curated manually to produce harmonized derived products (Read et al., 2017). Few examples of lake water quality data exist that harmonize both manually 40 sampled and autonomously sampled high-frequency data across key gradients in space and across decadal timescales.

A benchmark dataset for lake water quality that has well-resolved temporal data spanning multiple variables would be invaluable to both limnologists and computer scientists for simultaneously advancing both water quality modeling and KGML. Benchmark datasets are curated and cleaned datasets used in computation-heavy fields to test new operational methods and compare their performances (Peters et al., 2018). High-quality benchmark datasets are a significant effort to create (Sarkar et 45 al., 2020) but are of fundamental importance to the field of computer science (Li et al., 2024). These datasets are becoming more prevalent in the field of ecology (e.g., Weinstein et al., 2021; Schür et al., 2023). Ecological benchmark datasets are vital as environmental data, including water quality data, exhibit properties such as prevalent missing values and non-normal distributions (Helsel, 1987; Lim and Surbeck, 2011) that are not typically represented in machine learning benchmark datasets. Benchmark datasets exist within the field of hydrology (e.g., Addor et al., 2017; Demir et al., 2022) and some recent limnology 50 datasets advertise machine learning as a potential application (e.g., Spaulding et al., 2024), but benchmark datasets are rare in the field of limnology. This scarcity has caused some limnological studies to use non-limnological benchmark datasets to test their machine learning methods (e.g., Kadkhodazadeh and Farzin, 2021).

This paper introduces LakeBeD-US, a dataset of lake water quality time series and vertical profiles intended as a benchmark for comparative methodological analysis for water quality modeling. LakeBeD-US harmonizes water quality data from long-55 term water quality monitoring programs, including the North Temperate Lakes Long-Term Ecological Research program (NTL-LTER), National Ecological Observatory Network (NEON), Niwot Ridge Long-Term Ecological Research program (NWT-LTER), and the Carey Lab at Virginia Tech as part of the Virginia Reservoirs Long-Term Research in Environmental Biology



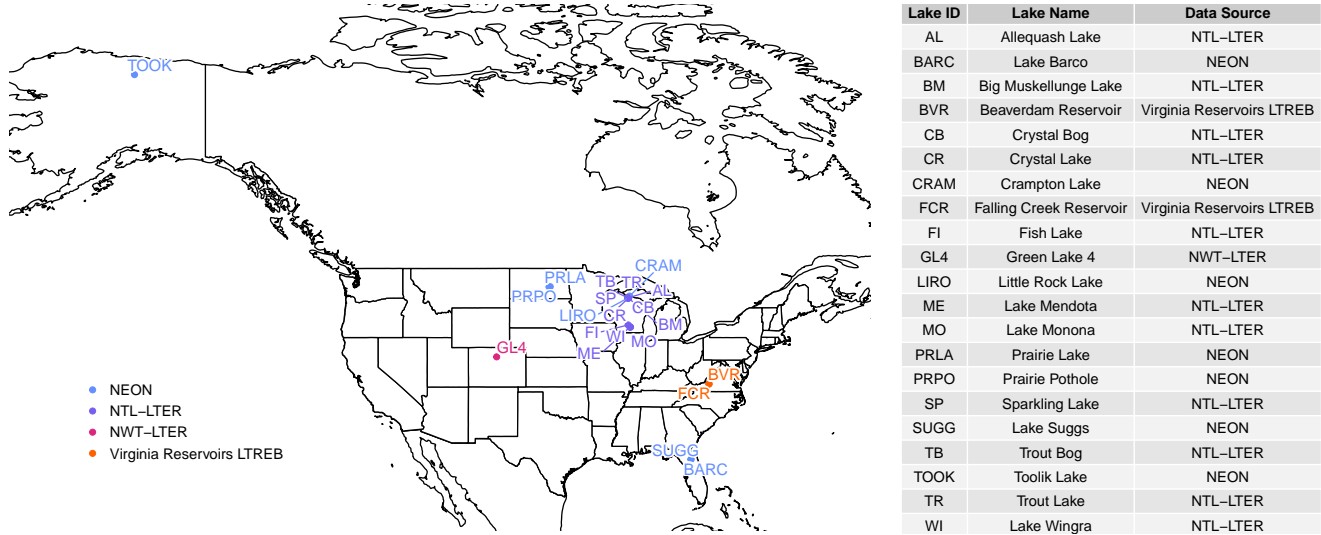

**Figure 1.** Locations and names of the 21 lakes included in LakeBeD-US. Lakes are monitored by the National Ecological Observatory Network (NEON, blue), North Temperate Lakes Long-Term Ecological Research program (NTL-LTER, purple), Niwot Ridge Long-Term Ecological Research program (NWT-LTER, pink), and the Carey Lab at Virginia Tech as part of the Virginia Reservoirs Long-Term Research in Environmental Biology (LTREB) site in collaboration with the Western Virginia Water Authority (orange).

(LTREB) site in collaboration with the Western Virginia Water Authority. To conform with the principles of FAIR (Findable, Accessible, Interoperable, and Reusable) data (Wilkinson et al., 2016), the data are accessible via digital object identifiers (DOIs), the contents are richly described in the metadata, and all provenance is documented for each data point. Data from 21 lakes are included. The group of lakes vary in size, geographic region, trophic status, and temporal coverage. LakeBeD-US is published in two forms, each with a unique DOI: LakeBeD-US: Ecology Edition is published in the Environmental Data Initiative repository (McAfee et al., 2024), which is a repository of primarily ecological data (Gries et al., 2023). LakeBeD-US: Computer Science Edition is published in the Hugging Face repository (Pradhan et al., 2024), which is used heavily by scientists developing and testing machine learning algorithms (Jain, 2022; Yang et al., 2024). Both versions are published as Apache Parquet files, a space-efficient and programming language-independent file type effective for storing time series data (Rangaraj et al., 2022). LakeBeD-US: Computer Science Edition is derived from LakeBeD-US: Ecology Edition with additional cleaning and reformatting described in Section 2 of this paper.

## 2 Dataset components and assembly

The goal of LakeBeD-US is to feature data from a collection of well-observed lakes that showcase the varied morphological, geographical, anthropological, and biological characteristics of environments across the United States. To do this, we leveraged data collected by prominent long-term monitoring programs. The NTL-LTER sampling strategy focuses on heterogeneous lakes within the state of Wisconsin (Magnuson et al., 2006). NEON samples lakes across the continent, capturing additional climactic



and land-use gradients (Goodman et al., 2015). Green Lake 4 from the NWT-LTER was chosen as a representative of alpine lakes in the dataset, as it has been monitored for many years (Bjarke et al., 2021). Falling Creek Reservoir and Beaverdam Reservoir represent managed drinking water supply reservoirs (Carey et al., 2024) which may exhibit unique characteristics as a result of their human influence. The degree to which the lakes in LakeBeD-US vary is discussed further in Section 3.

The LakeBeD-US dataset is presented in two formats: an Ecology Edition (McAfee et al., 2024) and a Computer Science Edition (Pradhan et al., 2024). LakeBeD-US: Ecology Edition is formatted to support analyses of lake water quality by the limnology community, while LakeBeD-US: Computer Science Edition is formatted for use with machine learning and KGML methods. The Ecology Edition is presented in a long format, with each water quality variable sharing columns such that variables of interest can be queried from the dataset using dplyr (Wickham et al., 2023) commands in R (R Core Team, 2023) and visualizing time series with common plotting tools like ggplot2 (Wickham, 2016) can be done efficiently. The Computer Science Edition is presented in a wide format where each water quality variable is presented in its own column, enabling their use as separate features in a machine learning model. More information about the two versions is presented in Table 1 and discussed further in Section 2.2.2.

## 2.1 LakeBeD-US: Ecology Edition

### 2.1.1 Source data harmonization

LakeBeD-US: Ecology Edition was assembled by downloading the source data to R (version 4.3.3, R Core Team, 2023) using the "EDIutils" (version 1.0.3, Smith, 2023) and "neonUtilities" (version 2.4.2, Lunch et al., 2024) packages. The data were harmonized using the Tidyverse suite of packages (version 2.0.0, Wickham et al., 2019) before being exported to Apache Parquet files with the "arrow" package (version 15.0.1, Richardson et al., 2024). The code to download and harmonize the data was written to search the source repository for the most updated version of the source data prior to harmonization. The specific version of the source data used is tracked in a separate table, listed in the code as the `provenance` object, that is manually checked for changes before further use of LakeBeD-US.

During harmonization into the LakeBeD-US: Ecology Edition format, all measurements of the same variable were converted into common units. The only exception to this is chlorophyll $a$, which comes in two types of units that are not directly comparable without additional analysis: Relative Fluorescence Units (RFU) and micrograms per liter ($\mu g\,L^{-1}$). Most of the source data related to nutrients or chemistry were already reported in either $\mu g\,L^{-1}$ or milligrams per liter ($mg\,L^{-1}$) which were straightforward to convert between. Data reported in molar units or microequivalents were converted to mass concentration units. Photosynthetically active radiation (PAR) data reported in lux units were converted to micromole per square meter per second ($\mu mol\,m^{-2}\,s^{-1}$) using the full sunlight conversion factor of 0.0185 (Thimijan and Heins, 1983).

### 2.1.2 Lake information table

The lake information table contains static attributes of the 21 lakes included in LakeBeD-US. These attributes include the monitoring institution, latitude, longitude, elevation above sea level, lake surface area, the mean and maximum depth, an



**Table 1.** Characteristics of the different formats of LakeBeD-US

| Ecology Edition | Computer Science Edition |
| --- | --- |
| – Long-format enables querying of the data by lake, variable, or quality flag with dplyr commands. Plot timeseries of multiple variables in ggplot2 with aesthetics arguments<br><br>– Included R script gives a tutorial on the use of Parquet files in R<br><br>– Complete data including sources and quality flags for manual data cleaning, allowing greater flexibility for users with limnology expertise. | – Wide-format enables straightforward machine-learning application where each variable acts as a feature<br><br>– Variables of different dimensionality (Static: vary by lake, 1D: vary through time, 2D: vary through time and depth) are partitioned to allow flexible model design<br><br>– Data are organized by lake for transfer learning experiments<br><br>– Duplicate observations are removed but all sources and quality flags are retained |

estimated hydrologic residence time, and any known manipulations of the lake performed by humans. These values were derived from published literature listing the attributes of each lake (listed in Appendix B2). Mean depth values were calculated based on available bathymetry information (Carey et al., 2022) when no values were reported in the literature. The hydrologic residence times listed are estimates based on the range of times the lake exhibits (Flanagan et al., 2009; Gerling et al., 2014).

An estimated hydrologic residence time was available for all lakes except for Fish Lake (Dane County, WI), a closed-basin lake with no surface water inflows or outflows. Elevations for each lake were obtained using the United States Geological Survey's (USGS) The National Map Bulk Point Query Service (United States Geological Survey, 2024). While there is uncertainty associated with the USGS 3D Elevation Program (Stoker and Miller, 2022), we found that elevation values captured the ecologically relevant variation and matched closely many published values for the 21 lakes in LakeBeD-US. Sources for each

specific attribute of a lake are listed as comments in the source code compiling the attributes, and listed collectively in the provenance metadata of LakeBeD-US: Ecology Edition.





### 2.1.3 High and low frequency observations tables

Observational data are compiled into two Parquet datasets: one representing data collected from a buoy-mounted sensor at a relatively high temporal frequency and the other collected by hand at a relatively low temporal frequency. The high- and low-frequency datasets use an identical format and can be easily merged if needed. However, there are many analytical considerations that differ between these temporal frequencies, so they are provided separately for LakeBeD-US: Ecology Edition. The low-frequency observation table includes a larger suite of variables and at a greater number of discrete depths along the water column.

Both the high- and low-frequency datasets are comprised of columns listing the source of a data point, date and time, the lake, depth, water quality variable, unit, observed value, and data flag. The data are provided in a long format for ease of querying the data by filtering with dplyr (Wickham et al., 2023) commands in R. All unit names were sourced from the QUDT (Quantities, Units, Dimensions and Types) ontology (FAIRsharing.org, 2022), with the exception of RFU which is not included in the ontology. Water quality variable names are defined in the metadata of both LakeBeD-US: Ecology Edition and LakeBeD-US: Computer Science Edition datasets.

### 2.1.4 Data flagging

Each of the original data sources (listed in Appendix B1) has data quality flagging systems that have been maintained for LakeBeD-US: Ecology Edition (Table A3). We documented all of the data quality flags in the original data sources and assigned each unique type of quality flag a number, aligning common types between each source. These numeric flags for LakeBeD-US are documented in the included Flag Guide table (Fig. 3; Table A3). As data go through the harmonization workflow to be included in LakeBeD-US: Ecology Edition, the original flag values are reassigned to align with the LakeBeD-US flag. There are 51 total unique flags among all of the data sources that were included in LakeBeD-US. Some of the data sources contain flags that are not defined in the metadata for those sources, in which case the data author was contacted and asked for a definition. Typically, these flags were errors in data entry and filtered out of LakeBeD-US. The exact depths at which some of the early buoy-mounted sensors were positioned were not documented and this institutional knowledge has been lost to time. Fortunately, documented protocols state that the sensors were mounted in the mixed surface layer of the lake. Thus, we have applied a depth value of 0.5 meters to those observations with the flag 52 attached.

## 2.2 LakeBeD-US: Computer Science Edition

### 2.2.1 Transformation from LakeBeD-US: Ecology Edition

LakeBeD-US: Computer Science Edition was transformed from the observational data and lake attribute information of LakeBeD-US: Ecology Edition. The original data was loaded with Python (version 3.12.4, Van Rossum and Drake, 2009) and transformed using pandas (version 2.2.2, The pandas development team, 2024; McKinney, 2010) and NumPy (version 2.1.1, Harris et al., 2020). Since the original data files were stored as Parquet files, additional dependencies fastparquet (ver-



sion 2024.5.0, Durant and Augsperger, 2024) and PyArrow (version 17.0.0, Apache Arrow Developers, 2024) were required for pandas. The transformation process consisted of five major components: flag imputation, data cleaning, variable renaming, deduplication and pivoting. The harmonization workflow is visualized in Fig. 2 and the steps taken in each component are outlined below:

1. **Flag Imputation:** Observations with missing values for `flag` were assumed to be accurate observations and imputed with a flag value of "0".

2. **Data Cleaning:** Some observations of 2D variables were assigned depth values of "-99" to indicate an integrated (i.e., taken from multiple depths simultaneously) observation. We omit those observations as they are not directly comparable to discrete-depth observations. It should be noted that several observations contain negative values for depth close to zero (on the order of $-10^{-3}$ to $-10^{-7}$ meters) but are correct observations. Such observations come from artificial reservoirs where the water level fluctuates greatly. As such, the depths for those observations need to be calculated from the reference surface level leading to some error in the depth measurement. It is permissible to round these values to zero if needed for simplification.

3. **Variable Renaming:** The `units` column of the observational data in LakeBeD-US: Ecology Edition was omitted in favor of listing the units in the metadata. However, chlorophyll *a* (`chla`) is reported in both RFU and µg L$^{-1}$ in the Ecology Edition. We separate this single variable with two units into `chla_rfu` and `chla_ugl` to distinguish between the two possible units of measurement for chlorophyll *a*.

4. **Deduplication:** The spatiotemporal nature of the data combined with flag values creates a bifurcation structure in the one-dimensional (1D, i.e., varying over time) variables and a trifurcation structure in the two-dimensional (2D, i.e., varying over time and depth) variables (see Section 2.2.2 for more information on variable types). A 1D observation can be indexed by `datetime` and `flag`, and a 2D observation can be indexed by `datetime`, `depth` and `flag`. Multiple observations could be present at a given index. We combine multiple observations at an index into a single observation by calculating the median.

5. **Pivoting:** LakeBeD-US: Ecology Edition is distributed in a "long" format where different variables are stored as a single column. This format was converted into a wide-format with tabular data where each variable has its own column. For 1D variables, `datetime` and `flag` were used as pivot indices, `variable` was used to denote the different resulting columns for the variables, and `observation` was used to populate the columns with values. Pivoting of the 2D data was performed identically except for the pivot indices where `datetime`, `depth` and `flag` were used.

### 2.2.2 File structure and components

LakeBeD-US: Computer Science Edition has a nested file structure as shown in Fig. 3. High- and low-frequency observational data are divided into two folders, each containing sub-folders for different lakes. Each lake's folder contains two tables: 1D variables and 2D variables. Static covariate information is stored in a spreadsheet (csv) file containing all lakes' information,





while the 1D and 2D variable data are stored as Apache Parquet files within the nested file structure. Static covariates are the lake attributes that generally remain constant over time, derived from the lake information table in LakeBeD-US: Ecology Edition. 1D variables have a temporal component but no depth information. Secchi depth is a standard 1D variable as it varies throughout time but is an attribute of the whole water column and thus cannot be sampled in a depth-discrete way. 2D variables vary by time and by depth, and each sample is depth-discrete. Quality flags are retained through the `flag` column of the 1D

and 2D variable tables.

Ecologists and computer scientists have different analytical approaches and thus different data structures are preferred when working with spatiotemporal data. Ecologists benefit from a long format because this file structure is well suited for aggregated statistics and complex data visualization. The long format also doesn't require the explicit storage of missing data. Computer scientists, on the other hand, benefit from a wide format due to its compatibility with machine learning workflows. At a high

level, machine learning algorithms implemented in popular libraries and frameworks (e.g., NumPy, PyTorch, scikit-learn and TensorFlow) expect data formatted like in the wide format. At a low level, specialized hardware like graphical processing units and tensor processing units, on which these libraries and frameworks are run, are optimized to operate on vector, matrix and tensor data structures. The wide format lends itself nicely to storage in these formats. Furthermore, wide-format data is often optimized for storage and querying in data systems to enhance computational performance when working with large datasets.

Lastly, having all variables in separate columns makes it easier to perform feature selection, engineering, and scaling, which are critical steps in preparing data for machine learning models.

## 2.3 Assessment and usage of data

To better understand the characteristics of LakeBeD-US, we showcase here the content of LakeBeD-US: Ecology Edition. Data were loaded into R using the "arrow" package and then queried using "dplyr" (version 1.1.4, Wickham et al., 2023). Visual-

ization made use of the "ggplot2" (version 3.5.1, Wickham, 2016), "ggrepel" (version 0.9.5, Slowikowski, 2024), "gridExtra" (version 2.3, Auguie, 2017), "cowplot" (version 1.1.3, Wilke, 2024), "maps" (version 3.4.2, Becker et al., 2023) and "mapdata" (version 2.3.1, Becker et al., 2022) libraries.

## 3 Ecology Edition: dataset characteristics

### 3.1 Spatial and temporal extent

While a majority of the lakes included in LakeBeD-US are north temperate lakes in the state of Wisconsin (Fig. 1), geographic variation is well represented in the dataset alongside other attributes. Toolik Lake is located in the North Slope Borough, Alaska and is the furthest northwest of any lake in the dataset (Fig. 1), representing an arctic system. In contrast, Lake Suggs and Lake Barco in Putman County, Florida represent the southeastern-most lakes and are located in a subtropical climate. Suggs and Barco also represent two of the polymictic lakes in the dataset alongside Prairie Lake (Stutsman County, ND), Prairie Pothole

(Stutsman County, ND), Lake Wingra (Dane County, WI), and Green Lake 4 (Boulder County, CO) (Preston et al., 2016;

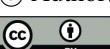

Earth System
Science
Data

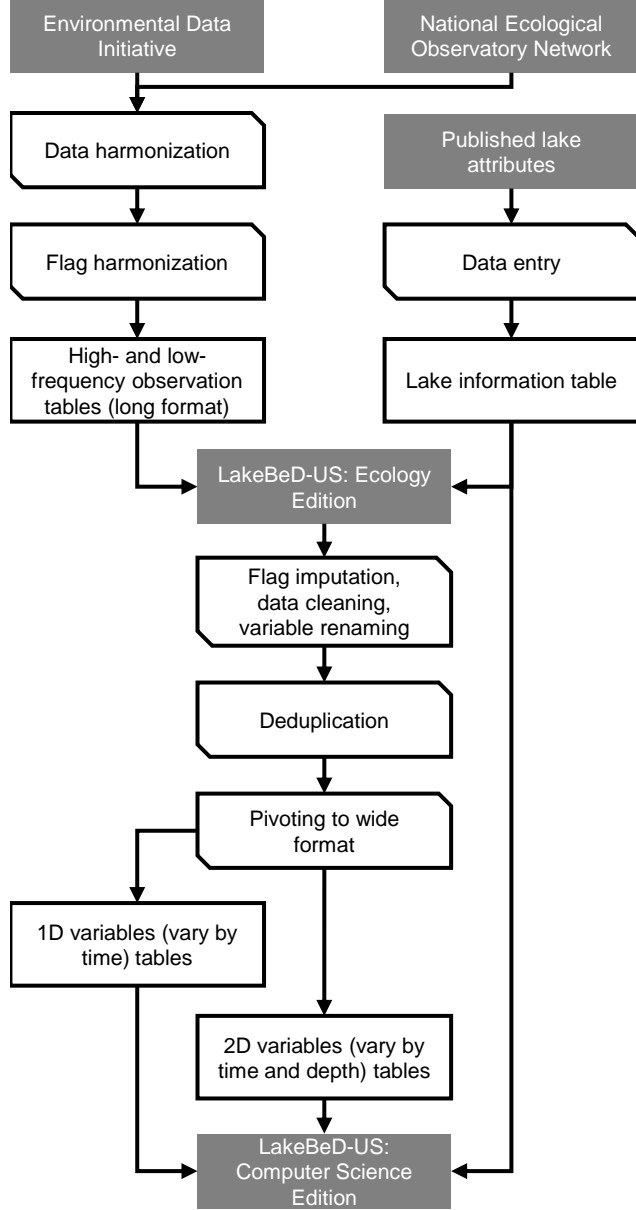

**Figure 2.** Harmonization workflow for LakeBeD-US. Boxes represent states of data and corner-snipped parallelograms represent processes. Grey boxes represent published products and sources of published products.

Thomas et al., 2023; Lottig and Dugan, 2024). All other lakes in the dataset are dimictic (Gerling et al., 2014; Thomas et al., 2023; Lottig and Dugan, 2024). Green Lake 4 represents the highest altitude lake in LakeBeD-US, with an elevation of over 3500 meters above sea level, a stark contrast to Lakes Suggs and Barco at approximately 27 meters (United States Geological Survey, 2024). Falling Creek Reservoir and Beaverdam Reservoir are drinking water reservoirs and thus experience a unique

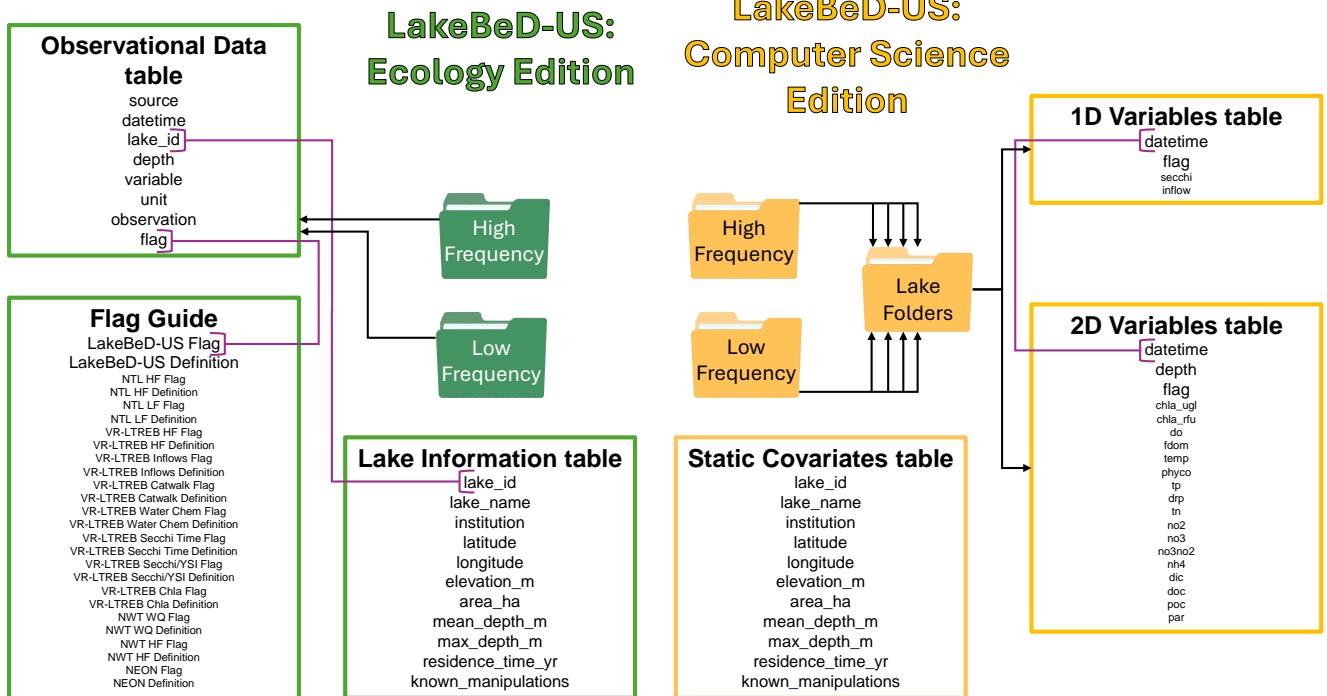

**Figure 3.** Structure of LakeBeD-US: Ecology Edition and LakeBeD-US: Computer Science Edition. Arrows indicate folder contents (e.g. LakeBeD-US: Ecology Edition contains High- and Low-Frequency folders that each contain an observational data table). Purple connectors indicate common columns by which to link tables. LakeBeD-US: Computer Science Edition contains High- and Low-Frequency Folders that each contain seperate folders seperating data from each lake. Each lake folder contains its own static covariate, 2D varables, and 1D variables tables.

set of human manipulations and impacts despite being in a relatively undisturbed forested watershed (Gerling et al., 2014). This provides a potential comparison to the lakes of the NTL-LTER in Wisconsin which have urban and agricultural catchments in the Madison area (Dane County) and relatively undisturbed forested catchments in Vilas County (Magnuson et al., 1997).

An overview of the temporal characteristics of observational data in LakeBeD-US is given in Table 2 and Fig. 4. The minimum time range of observed values for any lake in the dataset is five years, while seven NTL-LTER lakes have over 40

years of data. High-frequency data collection began between 2003 and 2006 for select NTL-LTER lakes, while the majority of high-frequency data collected comes from NEON starting in 2017. The Virginia Reservoirs LTREB and NWT-LTER high-frequency sensors were launched in 2013 and 2018, respectively. The longer-running high frequency programs measure fewer water quality variables (typically temperature, dissolved oxygen, and PAR) relative to the newer programs that have many additional variables including $NO_3$, fluorescent dissolved organic matter (fDOM), and chlorophyll *a*. Observations are also

not distributed evenly throughout the year. Observations from May through October, half the year during the ice-free season in



**Table 2.** Availability of observations by lake in LakeBeD-US: Ecology Edition. Counts for number of observations include all depths.

| Lake ID | Lake Name | Low Frequency Variables | Low Frequency Observations | Low Frequency Time Series | High Frequency Variables | High Frequency Observations | High Frequency Time Series |
|---|---|---|---|---|---|---|---|
| AL | Allequash Lake | 12 | 31546 | 1981-2022 | | | |
| BARC | Lake Barco | 11 | 2991 | 2014-2022 | 6 | 31836265 | 2017-2023 |
| BM | Big Muskellunge Lake | 12 | 56262 | 1981-2022 | | | |
| BVR | Beaverdam Reservoir | 11 | 11477 | 2013-2023 | 5 | 5186155 | 2013-2024 |
| CB | Crystal Bog | 12 | 16471 | 1981-2022 | 3 | 41784342 | 2005-2022 |
| CR | Crystal Lake | 12 | 59176 | 1981-2022 | | | |
| CRAM | Crampton Lake | 11 | 5877 | 2015-2022 | 6 | 16938998 | 2017-2023 |
| FCR | Falling Creek Reservoir | 11 | 23963 | 2013-2023 | 7 | 6641501 | 2013-2024 |
| FI | Fish Lake | 12 | 27045 | 1996-2022 | | | |
| GL4 | Green Lake 4 | 12 | 5331 | 1998-2023 | 4 | 1013065 | 2018-2023 |
| LIRO | Little Rock Lake | 11 | 2856 | 2017-2022 | 6 | 18890971 | 2017-2023 |
| ME | Lake Mendota | 13 | 37756 | 1995-2022 | 6 | 122268124 | 2006-2023 |
| MO | Lake Monona | 12 | 31294 | 1995-2022 | | | |
| PRLA | Prairie Lake | 11 | 2043 | 2014-2022 | 6 | 13331185 | 2017-2023 |
| PRPO | Prairie Pothole | 11 | 1586 | 2014-2022 | 6 | 12717778 | 2017-2023 |
| SP | Sparkling Lake | 12 | 55010 | 1981-2022 | 3 | 68603864 | 2004-2022 |
| SUGG | Lake Suggs | 11 | 1317 | 2014-2022 | 6 | 23744154 | 2017-2023 |
| TB | Trout Bog | 12 | 29337 | 1981-2022 | 3 | 77620538 | 2003-2022 |
| TOOK | Toolik Lake | 11 | 4365 | 2016-2022 | 6 | 6267990 | 2017-2023 |
| TR | Trout Lake | 12 | 77402 | 1981-2022 | 3 | 62471497 | 2004-2023 |
| WI | Lake Wingra | 12 | 8293 | 1996-2022 | | | |

temperate regions, make up 76.4% of the total number of observations in the dataset. However, there are observations present during winter months from lakes that do not freeze and from limited under-ice observations (e.g., Lottig, 2022).

The number of depths available for each variable in each lake at low- and high-frequencies are given by Tables 3 and 4. The number of depths sampled and at what intervals they are sampled is highly dependent on the water quality parameter being

measured. Among the manually sampled data, variables that can be measured via a sonde cast (e.g., water temperature and dissolved oxygen) are generally captured at a high spatial resolution with intervals of every 0.5 or 1 meter depending on the depth of the lake. Variables that are much more expensive or difficult to measure, such as dissolved nutrients, are generally measured at a much lower spatial resolution, sometimes only capturing the surface waters. Spatial resolution of high frequency



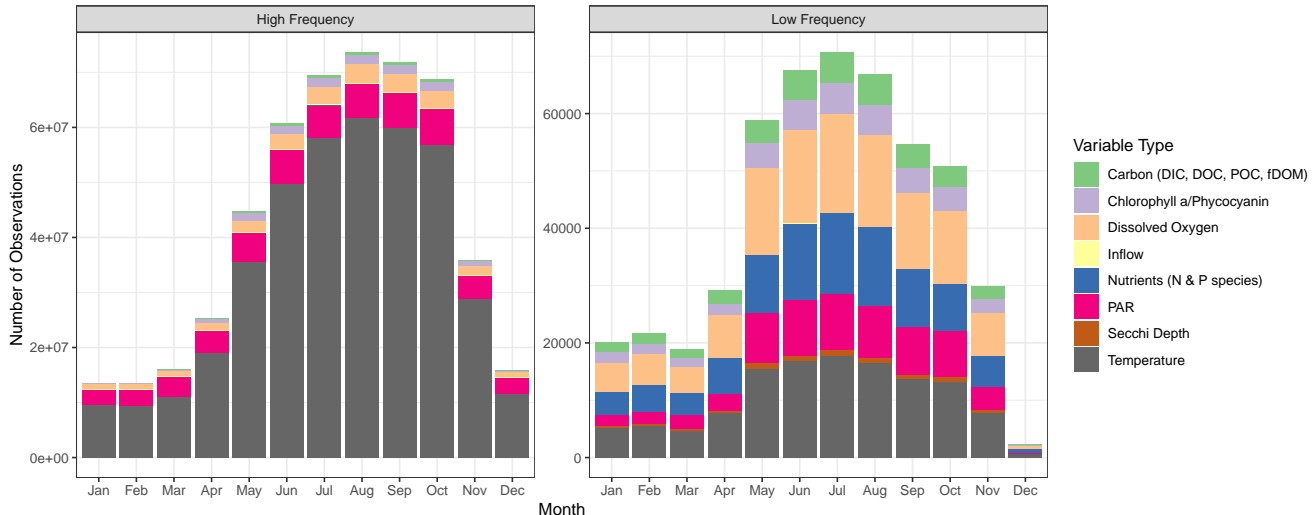

**Figure 4.** Temporal Distribution of observations in LakeBeD-US: Ecology Edition. Colors represent categories of variables.

**Table 3.** Number of depths, rounded to 0.5m, with more than 200 observations measured at a high frequency for each variable in each lake.

| Lake Name | Chl a | DO | fDOM | Inflow | $NO_3$ | PAR | Phycocyanin | Temp |
|---|---|---|---|---|---|---|---|---|
| BARC | 9 | 9 | 9 | | 1 | 2 | | 7 |
| BVR | 3 | 25 | 3 | | | 1 | | 25 |
| CB | | 3 | | | | 4 | | 7 |
| CRAM | 12 | 12 | 12 | | 1 | 2 | | 10 |
| FCR | 1 | 21 | 1 | 1 | | 1 | 1 | 21 |
| GL4 | 2 | 7 | | | | 2 | | 19 |
| LIRO | 12 | 12 | 12 | | 1 | 2 | | 10 |
| ME | 1 | 1 | 1 | | | 2 | 1 | 29 |
| PRLA | 4 | 4 | 4 | | 1 | 2 | | 5 |
| PRPO | 3 | 3 | 3 | | 1 | 2 | | 3 |
| SP | | 1 | | | | 1 | | 31 |
| SUGG | 5 | 5 | 5 | | 1 | 2 | | 3 |
| TB | | 3 | | | | 2 | | 17 |
| TOOK | 17 | 17 | 17 | | 1 | 2 | | 10 |
| TR | | 1 | | | | 1 | | 42 |

measurements varies by the monitoring institution, with some lakes focusing primarily on the surface waters while others
capture a greater number of depths.



**Table 4.** Number of depths, rounded to 0.5m, with more than 2 observations manually sampled for each variable in each lake. Secchi depth is 1D variable measured in all lakes that is not shown in this table.

| Lake ID | $NH_4$ | Chl a | DIC | DOC | DO | DRP | Inflow | $NO_3$ | $NO_2$ | $NO_2+NO_3$ | POC | PAR | TN | TP | Temp |
|---|---|---|---|---|---|---|---|---|---|---|---|---|---|---|---|
| AL | 10 | 11 | 6 | 6 | 9 | | | | 8 | 10 | | 17 | 6 | 9 | 11 |
| BARC | 1 | 1 | 1 | 1 | 13 | | | | 1 | 1 | | | 1 | 1 | 13 |
| BM | 18 | 17 | 14 | 14 | 21 | | | | 16 | 19 | | 22 | 14 | 15 | 22 |
| BVR | | 10 | 10 | 12 | 26 | 12 | | | | 12 | | 1 | 12 | 12 | 26 |
| CB | 5 | 4 | 2 | 2 | 6 | | | | 5 | 5 | | 7 | 2 | 3 | 6 |
| CR | 18 | 15 | 14 | 14 | 21 | | | | 17 | 19 | | 22 | 13 | 16 | 21 |
| CRAM | 1 | 1 | 1 | 1 | 36 | | | | 1 | 1 | | | 1 | 1 | 36 |
| FCR | | 6 | 10 | 13 | 22 | 13 | | | | 13 | | 1 | 13 | 13 | 22 |
| FI | 10 | 13 | 10 | 10 | 44 | 5 | | | | 10 | | 23 | 10 | 10 | 44 |
| GL4 | 5 | 5 | | 5 | 20 | | | 16 | | | 3 | 12 | 3 | 3 | 16 |
| LIRO | 1 | 1 | 1 | 1 | 18 | | | | 1 | 1 | | | 1 | 1 | 18 |
| ME | 14 | 8 | 13 | 15 | 49 | 12 | | | | 14 | 5 | 25 | 14 | 15 | 49 |
| MO | 11 | 8 | 12 | 12 | 45 | 11 | | | | 11 | | 21 | 11 | 12 | 45 |
| PRLA | 1 | 2 | 1 | 1 | 7 | | | | 1 | 1 | | | 1 | 1 | 7 |
| PRPO | 1 | 1 | 1 | 1 | 5 | | | | 1 | 1 | | | 1 | 1 | 6 |
| SP | 18 | 16 | 14 | 14 | 20 | | | | 18 | 20 | | 21 | 14 | 15 | 21 |
| SUGG | 1 | 2 | 1 | 1 | 4 | | | | 1 | 1 | | | 1 | 1 | 4 |
| TB | 10 | 9 | 8 | 8 | 9 | | | | 8 | 10 | | 15 | 8 | 8 | 10 |
| TOOK | 1 | 1 | 1 | 1 | 43 | | 1 | | 1 | 1 | | | 1 | 1 | 43 |
| TR | 24 | 32 | 18 | 18 | 37 | | | | 17 | 28 | | 30 | 18 | 20 | 37 |
| WI | 2 | 4 | 2 | 2 | 9 | 2 | | | | 2 | | 5 | 2 | 2 | 9 |

## 3.2 Water quality characteristics

The distributions of select lake attributes and water quality variables are given in Fig. 5 and Fig. 6. A wide range of lakes are present in LakeBeD-US in terms of surface area, depth, and indicators of trophic status. Of note, water quality variables often follow a non-normal distribution (Helsel, 1987; Lim and Surbeck, 2011), and LakeBeD-US is no exception (Fig. 6). This skewness is characteristic of environmental data, and should be considered by users of the dataset.





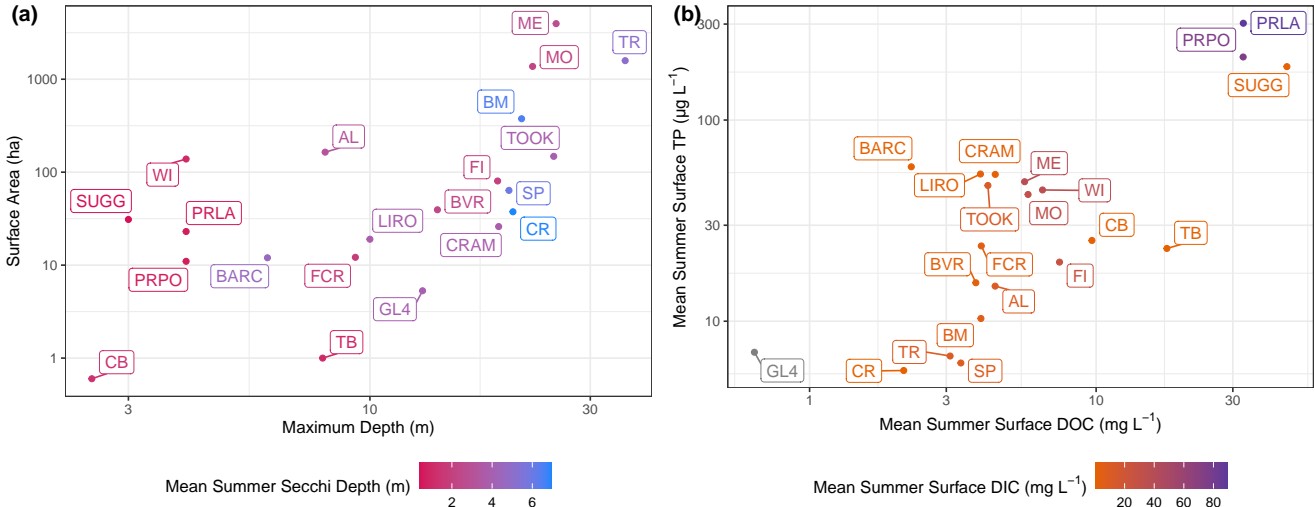

**Figure 5.** Lakes plotted within gradients of: a) Surface area in hectares compared to the maximum depth of the lake in meters. Points and labels are colored according to the mean summertime Secchi depth in meters. b) Mean summertime surface total phosphorus (TP) concentration in micrograms per liter compared to the mean summertime surface dissolved organic carbon (DOC) concentration in milligrams per liter. Points and labels are colored according to the mean summertime surface concentration of dissolved inorganic carbon (DIC) in milligrams per liter. Green Lake 4 (GL4) has no observational data for DIC. The surface is defined as the minimum depth sampled for the given time period, which was always within the surface mixed layer of the lake. Summer is defined as the months of June, July, and August.

## 4 Benchmark task

### 4.1 Computer Science Edition benchmark

We demonstrate the utility and applicability of LakeBeD-US: Computer Science Edition for the machine learning task of multivariate time series prediction. We predict the daily median dissolved oxygen concentration (`do`) and daily median water temperature (`temp`) at Lake Mendota using data from June 2006 to November 2023.

### 4.1.1 Data selection

LakeBeD-US provides two datasets for Lake Mendota, low-frequency and high-frequency; both were considered in this benchmark. As indicated earlier, every observation is provided with a flag code to indicate any erroneous measurements. Observations from the low- and high-frequency datasets with the flag codes indicated in Table 5 were selected for use in the benchmarking task.

While LakeBeD-US features spatiotemporal data, we considered data across a single depth of Lake Mendota to simplify the benchmark. This required considering the percentage of missing values at each depth that the low- and high-frequency datasets reported. For the high-frequency data, 100% of observations for all features, with the exception of water temperature, were





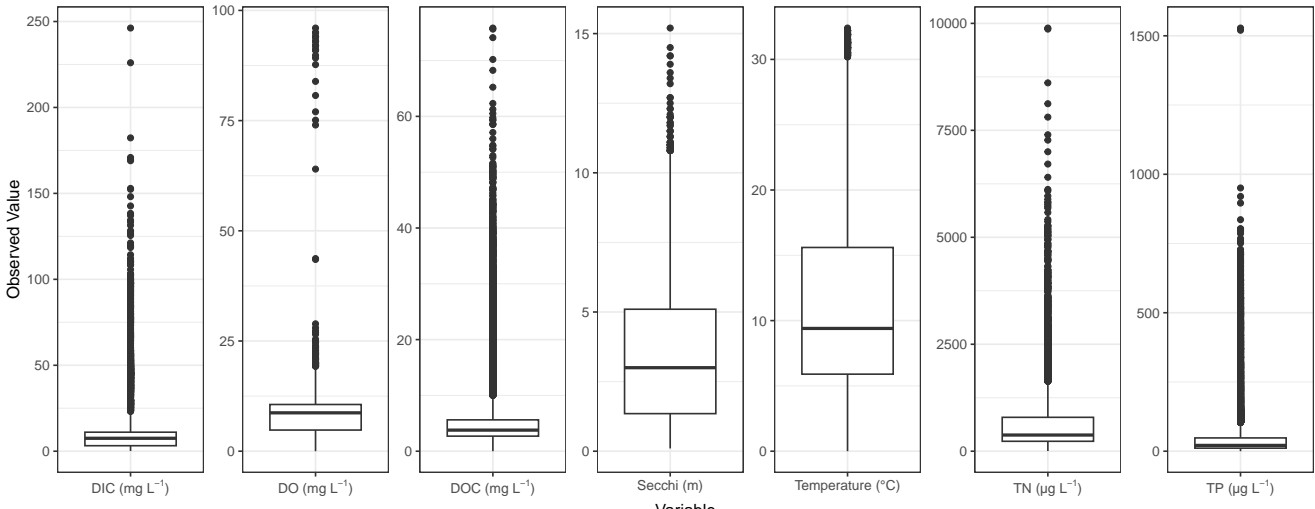

**Figure 6.** Distributions of observed values for dissolved inorganic carbon (DIC), dissolved oxygen concentration (DO), dissolved organic carbon (DOC), Secchi depth, temperature, total nitrogen concentration (TN), and total phosphorus concentration (TP) across all observations, after QA/QC, of all lakes in LakeBeD-US: Ecology Edition.

missing at all depths except 1.0 meters. Similarly the low-frequency data reported large percentages (>85-90%) of missing values for all variables across all depths. Among all variables reported in the datasets, we selected chlorophyll *a* (`chla_rfu`), photosynthetic active radiation (`par`) and phycocyanin (`phyco`) from the datasets to be used as covariates due to the high number of observations present for these variables in the high-frequency data.

### 4.1.2 Data wrangling

The following steps were taken to prepare the data for modeling:

1. **Timescale Standardization:** The timescales of the low and high-frequency datasets were discontinuous (e.g., the time series might have skipped from January 22nd to February 1st, thereby omitting January 23rd to 31st). We created two uniform timescales with no discontinuities at the resolution that was permitted by each dataset (i.e., daily resolution for the low-frequency data and minutely resolution for the high-frequency data) from the earliest to the most recent datetimes in both datasets. The observations that were present in the datasets were merged with their corresponding new timescale. The new low-frequency dataset's timescale spanned from May 9th, 1995 to November 1st, 2022 while the new high-frequency dataset's timescale spanned from June 28th, 2006 02:31:00 (02:31:00 AM) to November 19th, 2023 15:26:00 (3:26:00 PM). This step was critical to providing a more accurate value for the percentage of missing data.

2. **Data Harmonization:** To mitigate the issue of high percentages of missing observations, the low- and high-frequency datasets were merged into a single dataset. Since the low-frequency dataset begins in 1995, as opposed to the high-





**Table 5.** Acceptable flag codes for selected data used in the LakeBeD-US: Computer Science Edition benchmark. Not all the flag codes listed are relevant to the variables used in the benchmark task, but all are flags we would consider acceptable within QA/QC steps for most tasks using LakeBeD-US.

| Flag | Description |
|------|-------------|
| 0 | No flag |
| 5 | Average of duplicate analyses |
| 10 | Nonstandard methods |
| 19 | Value below detection limit; set to zero |
| 23 | Negative value set to zero |
| 25 | Sensor was off during part of the averaged period |
| 32 | Date is accurate but time is inaccurate |
| 43 | Sample run using NPOC (non-purgeable organic carbon) method due to high inorganic carbon values |
| 47 | Flagged with no explanation |
| 51 | Secchi depth hit bottom (calculated for NEON Lakes only) |
| 52 | Unknown depth near surface. Labeled as 0.5m |

frequency dataset which begins in 2006, the resulting harmonized dataset had an even larger percentage of missing values. From this merged dataset, we selected only the observations recorded since the start of the high-frequency dataset to minimize the amount of imputation that would be required.

3. **Downsampling and Aggregation:** The harmonized dataset was downsampled to a daily resolution by combining observations using a median aggregation function, resulting in a daily median time series.

4. **Splitting and Sliding Window Sampling:** The data was split 80%-10%-10% chronologically into a training-validation-testing split. The training split was standardized (Z-score normalized) and the standardization parameters were applied to the validation and testing splits. After standardization, windowed samples were generated for each split. A windowed sample consists of 21 days of observations of all features (`chla_rfu`, `par`, `phyco`, `do`, `temp`) as inputs, referred to as a "lookback window", and the subsequent 14 days of `do` and `temp` as targets, referred to as a "horizon window". For example, if we considered the observations of all features from January 1st to January 21st as the lookback window, the `do` and `temp` observations from January 22nd to February 4th would be the horizon window. The subsequent sample would be formed by "sliding the (lookback and horizon) window" by one day into the future (i.e. the second sample's lookback window would span January 2nd to January 22nd and the horizon window would span from January 23rd to February 5th). The sampling was carried out such that the horizon window of the last sample would not extend farther



**Table 6.** Start and end dates of each lookback and horizon window for the first and last samples in each split.

| Split | Sample | Window | Start Date | End Date |
|-------|--------|--------|------------|----------|
| Training | First | Lookback | 2006-06-28 | 2006-07-18 |
| | | Horizon | 2006-07-19 | 2006-08-01 |
| | Last | Lookback | 2020-04-23 | 2020-05-13 |
| | | Horizon | 2020-05-14 | 2020-05-27 |
| Validation | First | Lookback | 2020-05-28 | 2020-06-17 |
| | | Horizon | 2020-06-18 | 2020-07-01 |
| | Last | Lookback | 2022-01-18 | 2022-02-07 |
| | | Horizon | 2022-02-08 | 2022-02-21 |
| Testing | First | Lookback | 2022-02-22 | 2022-03-14 |
| | | Horizon | 2022-03-15 | 2022-03-28 |
| | Last | Lookback | 2023-10-16 | 2023-11-05 |
| | | Horizon | 2023-11-06 | 2023-11-19 |

**Table 7.** Percentage of missing values per variable in each split.

| Split | Percentage of Missing Values | | | | |
|-------|----------|--------|--------|--------|--------|
| | chla_rfu | par | phyco | do | temp |
| Training | 49.636 | 81.527 | 51.977 | 54.131 | 43.97 |
| Validation | 39.055 | 35.433 | 3.850 | 38.898 | 34.646 |
| Testing | 32.390 | 42.610 | 32.390 | 30.818 | 32.075 |

than the end of each respective split to avoid data leakage between splits. The start and end dates of the lookback and horizon window of the first and last sample in each split are given in Table 6.

5. **Imputation:** Prior to windowed sampling, the percentages of missing values for each split were calculated. These values are listed in Table 7.

The missing values in the input lookback windows for each split were imputed using the Self-Attention-based Imputation for Time Series (SAITS) method (Du et al., 2023). Traditional imputation techniques, such as spline interpolation and $k$-nearest neighbors, often rely on assumptions about simple relationships between adjacent data points. In contrast, SAITS leverages a self-attention mechanism to identify and emphasize relevant information across the entire dataset, even when pertinent data points are temporally distant. This approach allows SAITS to effectively capture complex temporal patterns and inter-variable relationships. During training, SAITS introduces artificial missing values into the dataset and attempts to impute them. By minimizing the discrepancy between its imputations and the original values,





**Table 8.** SAITS imputation model hyperparameters.

| Hyperparameter | Value |
|---|---|
| Sequence length | 21 |
| Number of features | 5 |
| Number of layers in the 1st and 2nd DMSA blocks | 2 |
| Model embedding dimensionality | 256 |
| Multi-head DMSA mechanism headcount | 4 |
| DMSA mechanism key and query dimensionality | 64 |
| DMSA mechanism value dimensionality | 64 |
| Feed-forward layer dimensionality | 128 |
| Fully-connected layer dropout rate | 0.1 |
| Epochs | 50 |
| Batch size | 32 |

SAITS learns to accurately reconstruct missing data, resulting in more reliable and comprehensive datasets for analysis.

A SAITS model was trained on the windowed samples from the training split using the hyperparameters specified in Table 8 and applied on the input lookback windows of the training, validation and testing splits. Since no ground truth for the dataset was present, the quality of the imputation could not be empirically measured and instead was inferred through the predictive skill of the model. The target horizon windows were not imputed because it would have been difficult to identify if strong performance of the model was a result of a good model or an overly simplistic imputation (e.g., a simple horizontal line).

### 4.1.3 Modeling

We outline the components of our modeling process for the benchmark below. All modeling was done in PyTorch (Paszke et al., 2019).

– **Model Architecture:** A sequence-to-sequence (seq2seq) long short-term memory recurrent neural network (LSTM-RNN) was constructed to predict dissolved oxygen concentration and water temperature. Seq2seq modeling arose from the field of natural language processing, specifically neural machine translation (NMT; Cho et al., 2014; Sutskever et al., 2014). In NMT, given an input sentence in one language, we wish to translate the sentence to another language, using a neural network, such that the translation has semantic meaning and obeys syntax of the target language. Observations in a time series, like words in a sentence, have an inherent temporal ordering. Thus, the problem of time series prediction conveniently lends itself to this modeling paradigm.





A seq2seq model follows an autoencoder architecture, comprising two main components: an encoder and a decoder. The encoder is built on an LSTM-RNN. It processes the input data in a sequential manner, mapping the input to a high-dimensional vector, called a "hidden state", at each timestep of the input. This hidden state exists in a latent feature space (also referred to as embedding space) which can abstractly be thought of as a summary of the input sequence up to that moment in time. When the encoder has encoded the final timestep of the input sequence into a hidden state, the final hidden state vector now contains a summary of the entire input time series. This final hidden state is referred to as a "context vector" that encapsulates the critical information of the sequence in a compressed form. The decoder, another LSTM-RNN, uses this context vector as a foundation to generate the desired target sequence. Operating in an autoregressive manner, the decoder predicts each timestep in the future sequence, feeding each prediction back as input to inform the next. This autoregressive process continues until the full sequence in the prediction window is generated.

– **Training Strategy**

– **Cost Function:** The parameters of the model were trained by minimizing the the root mean square error (RMSE) between the predicted target horizon window and the observed horizon window. Since the target horizon windows in each sample were not imputed, a masked loss computation was employed. In situations where the observed horizon window contained missing observations, the error was only computed between observations that were jointly present in the prediction and the observed horizon window. If the horizon window contained no observations, then the sample was omitted from the error computation. The RMSE cost function was minimized using the Adaptive Moment Estimation (AdaM) optimizer and a "Reduce Learning Rate on Plateau" learning rate scheduler. Learning rate scheduling is a technique to adaptively adjust the learning rate during training based on the model's performance on the validation split. The core idea is to reduce the learning rate when progress stalls, helping the model to escape saddle points or local minima in the cost landscape, thereby potentially achieving a final better result.

– **Regularization:** We leveraged early stopping and weight decay regularization. Early stopping is a regularization technique that mitigates overfitting of the model by monitoring the performance on the validation split. If the validation cost starts to increase over time, the model halts the training process. Weight decay is a regularization technique that operates by subtracting a fraction of the previous weights when updating the weights during training, effectively making the weights smaller over time. This subtraction of a portion of the existing weights ensures that during each iteration of training, the model's parameters are nudged towards smaller values.

– **Hyperparameter Selection:** Model architecture and learning hyperparameters were optimally chosen using the "Tree-structured Parzen Estimator" algorithm in the Optuna library by minimizing the validation cost over 50 trials (Akiba et al., 2019). The final hyperparameters are given in Table 9.





**Table 9.** Final model architecture and learning hyperparameters.

| Hyperparameter | Value |
|---|---|
| Hidden state dimensionality | 8 |
| Encoder recurrent layers | 1 |
| Decoder recurrent layers | 1 |
| Batch size | 32 |
| Epochs | 100 |
| Initial Learning Rate | $8.799 \times 10^{-4}$ |
| Learning rate decay factor | 0.1 |
| Learning rate scheduler patience | 3 epochs |
| Learning rate scheduler threshold | $1 \times 10^{-4}$ |
| Weight Decay | $3.0187 \times 10^{-4}$ |
| Early stopping patience | 5 epochs |

**Table 10.** Model performance on each data split as measured with standardized RMSE across five trials.

| Split | RMSE |
|---|---|
| Training | $0.44 \pm 0.02$ |
| Validation | $0.42 \pm 0.02$ |
| Testing | $0.36 \pm 0.01$ |

### 4.1.4 Results

The learning curve shown in Fig. 7a shows the performance of the model on the training and validation splits at each epoch in the training process while the learning rate schedule in Fig. 7b shows the reduction in the learning rate of the model until convergence. The final standardized RMSE of the model on each data split is presented in Table 10.

The predictions for dissolved oxygen concentration and water temperature for the training, validation and testing splits are shown in Fig. 8. For a given split, after the 21st day in the window, a 14-day ahead series of predictions is generated on each

350 day. This results in multiple, potentially up to 14, overlapping predictions for a single day. We consolidated these overlapping predictions by computing the median predicted value for each day across all predictions. This yields a single, continuous series of predictions for an entire split's timeline. The predictions shown in Fig. 8 were obtained by continuous predictions from each trial in each split. The confidence interval was generated by taking the minimum and maximum values for each datetime across each continuous series of predictions. Additionally, we report an unstandardized RMSE between the continuous predictions

from each trial and the observed dissolved oxygen concentration and water temperature across the entire timeline of each split in Table 11.



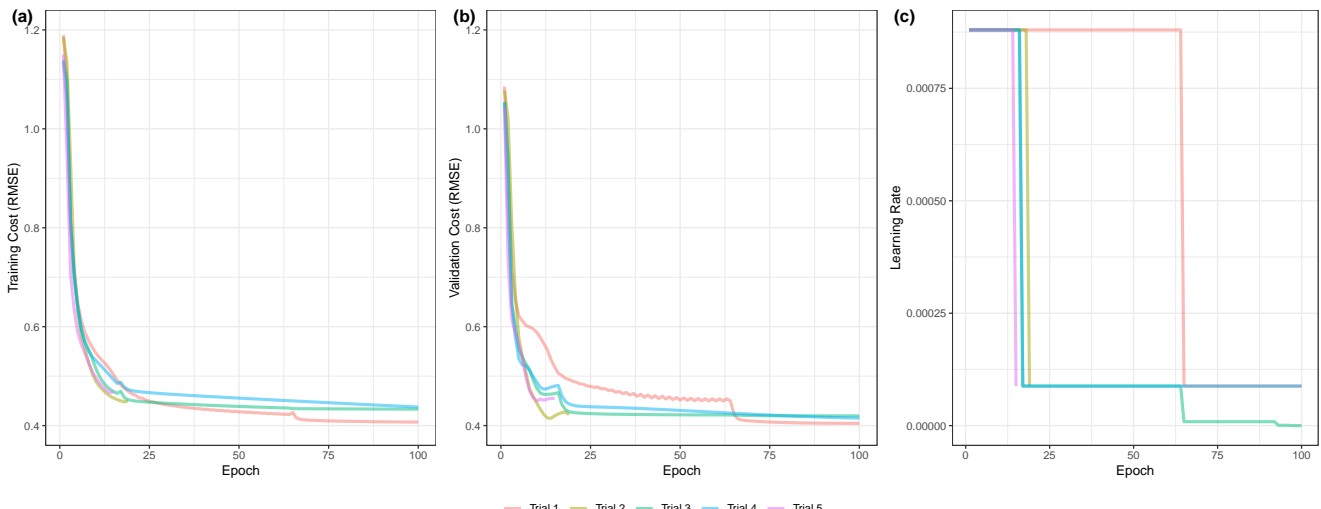

**Figure 7.** Training cost, validation cost, and learning rate of machine learning benchmark. The cost, a measure of model error, for the training (a) and validation (b) splits is shown as a function of the number of epochs. The learning rate, or amount of change between iterations of the model in response to error, is shown (c). Five trials are shown in different colors.

**Table 11.** Mean masked, unstandardized RMSE and standard deviation between the continuous timeseries predictions and observed values for each split across all trials. Dissolved oxygen (`do`) is reported in milligrams per liter and water temperature (`temp`) is reported in degrees Celsius.

| Variable | RMSE | | |
|---|---|---|---|
| | Training | Validation | Testing |
| `do` | $1.59 \pm 0.05$ | $1.61 \pm 0.19$ | $1.40 \pm 0.09$ |
| `temp` | $3.44 \pm 0.08$ | $3.65 \pm 0.22$ | $3.78 \pm 0.12$ |

### 4.1.5 Benchmark task discussion

With our benchmark task, we showcase the applicability of LakeBeD-US to multivariate timeseries prediction of two water quality variables. Our machine learning model performed comparably to existing process-based models for the purpose of predicting dissolved oxygen concentration. In predicting the dissolved oxygen concentration of the surface of Lake Mendota, an iteration of the GLM-AED2 model (Hipsey et al., 2019) calibrated by Ladwig et al. (2021) reported a RMSE of $2.77 \, \mathrm{mg \, L^{-1}}$ and a model constructed by Hanson et al. (2023) reported an RMSE of $1.45 \, \mathrm{mg \, L^{-1}}$. Our model predicted dissolved oxygen in the testing dataset with an RMSE of $1.40 \, \mathrm{mg \, L^{-1}}$ (Table 11), which is comparable to the aforementioned process-based water quality models. When predicting temperature, our model did not perform as well as the process-based models that reported RMSE values in the range of 1.30°C, less then half our machine-learning model's error (Table 11). While the predictions

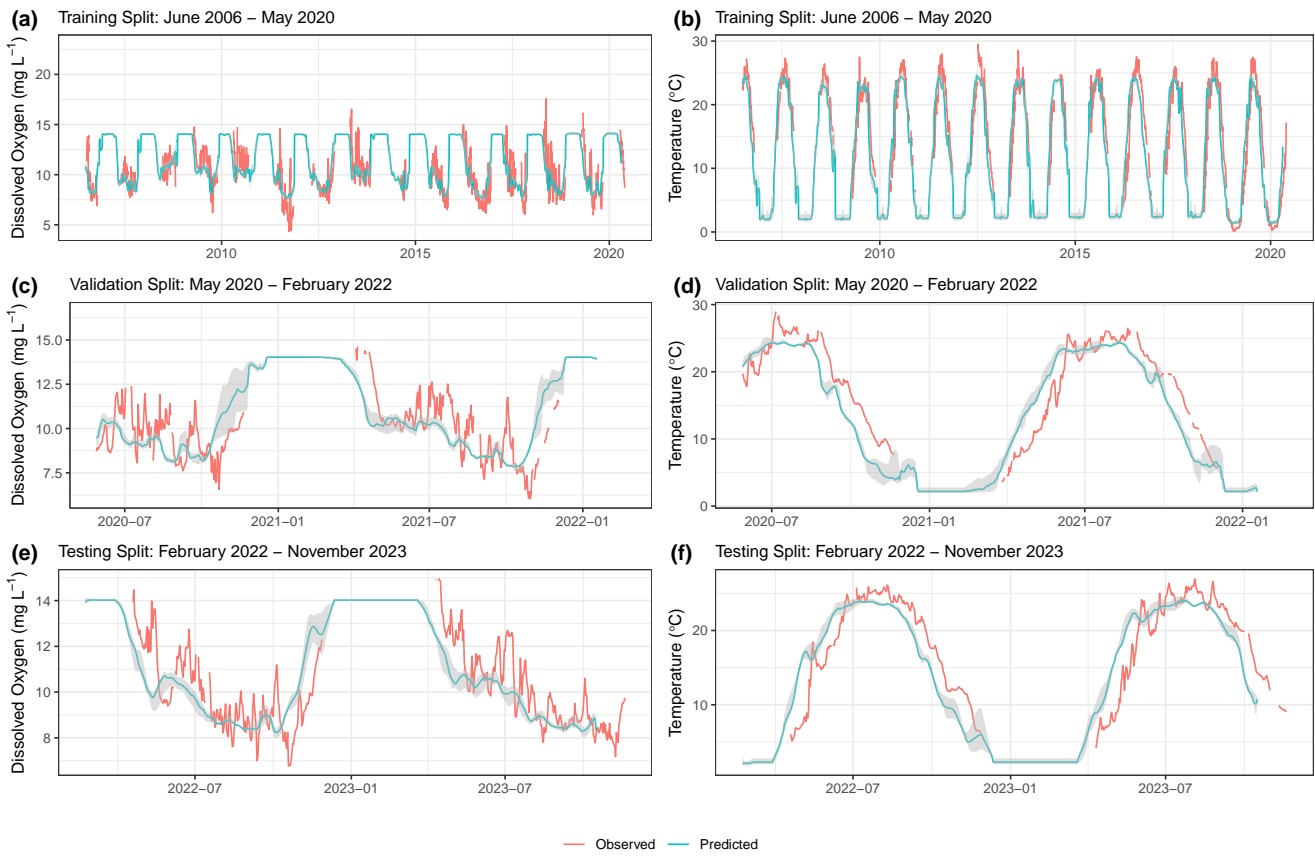

**Figure 8.** Machine learning model predictions of surface water dissolved oxygen and temperature. Observed (red) and predicted (blue) dissolved oxygen (a, c, e) and temperature (b, d, f) are shown. Training (a, b), validation (c, d), and testing (e, f) splits are shown. The grey shaded areas represent the confidence intervals.

generated by our model have room for improvement, they showcase that LakeBeD-US: Computer Science Edition can be used to create water quality models. Water temperature in particular is a variable that has been shown to be a useful tool for comparison of model performance in ecological tasks (Read et al., 2019).

## 5 Discussion

In this paper, we introduce LakeBeD-US: a dataset designed to foster the advancement of machine learning technologies in ecological applications. By combining spatially and temporally extensive datasets across a variety of scales, we offer a dataset that can be used in benchmarking tasks that address the scales of variability in the drivers of lake water quality. LakeBeD-US is compatible with ecological analysis, novel computer science methodologies, or both through the interdisciplinary paradigm of KGML.





## 5.1 Limitations and considerations for use of LakeBeD-US

LakeBeD-US is not a representative sample of the water quality gradients found in lakes of the world (Verpoorter et al., 2014; Messager et al., 2016). Limnological sampling efforts tend to favor large, easily accessible lakes that are used more frequently for their natural resources, and most samples are taken during the ice-free season (Stanley et al., 2019). A majority of the observations in LakeBeD-US represent the ice-free season (Fig. 4) and 13 of the 21 lakes included in the dataset are located within the state of Wisconsin (Fig. 1), with the most well-observed lake being the large, eutrophic, and heavily utilized Lake Mendota (Table 2). Despite this, LakeBeD-US still captures a variety of lake characteristics and geographic locations that enable users to investigate those attributes' relationship with water quality dynamics (Fig. 5).

LakeBeD-US does not account for methodological or equipment differences among its datasets. Sensors and laboratory procedures change over time and between monitoring institutions, which is information not present in LakeBeD-US, but is present in the source datasets (listed in Appendix B1). The harmonization procedure of LakeBeD-US assumes the accuracy and precision of the observed value of the source data, excepting any quality flags that have been applied to the data. Potential methodological differences should be investigated when encountering any unexplained changes in water quality trends present in LakeBeD-US.

The observed water quality variables exhibit a heavy skewness that is common among of data of its kind (Fig. 6; Helsel, 1987; Lim and Surbeck, 2011). Considerations should be taken when analyzing or using this data to limit the effect of this skewness, as omitting flagged values or outliers may not be enough (Virro et al., 2021). Transformations may need to be applied to the data before use, such as the standardization applied in the benchmark task of this paper.

Unlike many benchmark datasets, LakeBeD-US contains numerous missing values. This is a problem typical of environmental data. Fortunately, the handling of missing values in environmental data by machine learning algorithms is an active area of research (Rodríguez et al., 2021), and LakeBeD-US can act as a testing ground for developing novel methods.

## 5.2 LakeBeD-US as a machine learning benchmark

The benchmarking task in Section 4 is a straightforward example of how machine learning can be applied to lake water quality prediction using LakeBeD-US: Computer Science Edition. The machine learning model performed comparably to many existing process-based models when predicting dissolved oxygen concentration and temperature in Lake Mendota's surface waters (Section 4.1.5; Hanson et al., 2023; Ladwig et al., 2021). This showcases the applicability of machine learning to ecological problems, and the error in the model showcases the utility of LakeBeD-US as a benchmark dataset. Machine learning algorithms other than the LSTM-RNN used here may have a different performance for this task, an understanding of which is a vital part of the model selection process in ecological studies. The variety of lakes in LakeBeD-US enables future studies to investigate the performance of machine learning, mechanistic, and hybrid knowledge-guided machine learning models when making predictions across multiple lakes, trophic statuses, or temporal frequencies.

LakeBeD-US was assembled as part of an effort to advance the science of knowledge-guided machine learning (KGML) in ecological applications. There are many potential uses of the dataset for investigating water quality dynamics using these





techniques. Transfer learning is the use of a machine learning model trained on a number of source tasks applied to a new target task with limited data (Karpatne et al., 2024), which is a method that has been applied to lakes (Willard et al., 2021).

LakeBeD-US features a varied selection of lakes, making it suitable for the application of transfer learning methods for lake systems. Building upon this idea of transfer learning, there has been recent advancement in the application of foundation models to environmental data, which can be pre-trained on a broad, heterogeneous dataset and then fine-tuned on a more specific dataset to a given task (Lacoste et al., 2023; Nguyen et al., 2023; Karpatne et al., 2024). LakeBeD-US may prove useful in the application of foundation models, or other KGML methods (e.g., modular compositional learning; Ladwig et al.,

2024) to water quality.

## 5.3 Potential for the expansion of LakeBeD-US

While the number of lakes in LakeBeD-US is modest relative to national- or global-scale studies (e.g., Soranno et al., 2017; Solomon et al., 2013), the frequency and duration of its data provide unique opportunities for expanding scientific understanding of aquatic ecosystem dynamics. Ensuring that long-term water quality datasets meet the rigorous requirements for

LakeBeD-US requires working with the scientists and organizations who collect the data. This will have the added benefit of involving more lake ecologists in water quality modeling endeavors (Hanson et al., 2016). The Global Lake Ecological Observatory Network (GLEON) is an example of this type of community involvement in data collection, harmonization, and analysis (Weathers et al., 2013; Hamilton et al., 2015).

   Updates to include more data, more lakes, and more water quality variables are possible and collaboration in the creation of

new additions to LakeBeD-US is encouraged. The data provenance and versioning tools of the Environmental Data Initiative and Hugging Face repositories make it possible for specific versions of both LakeBeD-US: Ecology Edition and LakeBeD-US: Computer Science Edition to be referenced in future studies. The source code to harmonize LakeBeD-US: Ecology Edition searches the source repository for the latest release of the source data, enabling new updates to the existing sources to be integrated seamlessly, as long as major format changes in the source data do not occur. Adding new data sources to LakeBeD-

US: Ecology Edition is possible, requiring that a new R script to download and harmonize the observational data be written, and the Data_Controller.R and Lake_Info.R files in the source code be updated accordingly. LakeBeD-US: Ecology Edition's use of the Parquet format allows for additions to the dataset without having to rewrite the entirety of the dataset's files. LakeBeD-US: Computer Science Edition is created dynamically based on the content of LakeBeD-US: Ecology Edition, allowing for parity between the two versions. Stewards of long-term water quality monitoring data are encouraged to become contributors

to LakeBeD-US through the creation of modular additions to the Parquet dataset. These modules would emulate the design of LakeBeD-US: Ecology Edition, namely the R scripts to format the data and write it to Parquet files, meaning that users of the data could seamlessly add the contents of any community-made module to the base LakeBeD-US: Ecology Edition data. There would then be potential for integration of the data modules into the base LakeBeD-US in future revisions if that collaboration is desired.



## 6 Conclusions


LakeBeD-US is a dataset of lake water quality observations combining high- and low- frequency observations from 21 lakes across the United States collected by different monitoring institutions for the intention of AI benchmarking. This dataset is one of the first of its kind to capture water quality at a high spatial and temporal resolution in the selected lakes and be available in formats conducive to both ecological analyses and novel computer and data science approaches. As a benchmark dataset,


LakeBeD-US was designed to be used to advance the science of knowledge-guided machine learning and foster collaboration between ecologists and computer scientists.

There are many planned and potential uses for LakeBeD-US. As a benchmark dataset designed with machine learning in mind, LakeBeD-US offers an opportunity to test and compare new machine learning methodologies in an ecological context. Aspects of LakeBeD-US such as data skewness and missing values are prevalent in environmental data and this dataset offers


opportunities for the scientific community to investigate methods for mitigating these issues for machine learning models. This collection of data is also valuable for the investigation of water quality dynamics using statistical or mechanistic models. These advances in water quality modeling, prediction, and forecasting are vital in creating a greater understanding of aquatic systems and informing more thoughtful utilization of aquatic resources.

*Code and data availability.* LakeBeD-US: Ecology Edition is available in the Environmental Data Initiative repository (https://doi.org/10.


6073/pasta/c56a204a65483790f6277de4896d7140; McAfee et al., 2024). LakeBeD-US: Computer Science Edition is available in the Hugging Face repository (http://doi.org/10.57967/hf/3771; Pradhan et al., 2024).

**Appendix A: Table and Variable Metadata**





**Table A1.** LakeBeD-US: Ecology Edition Lake Information table metadata

| Column | Description | Data type |
| --- | --- | --- |
| lake_id | Identifier for a lake, common with the observational data | String |
| lake_name | Common name for a lake | String |
| institution | Monitoring institution responsible for collecting data on a lake. | String |
| latitude | Latitude of the deep hole of a lake in decimal degrees | Double |
| longitude | Longitude of the deep hole of a lake in decimal degrees | Double |
| elevation_m | Elevation of the lake in meters, determined by the USGS The National Map Bulk Point Query Service | Double |
| area_ha | Surface area of the lake in hectares | Double |
| mean_depth_m | Mean depth of the lake in meters, calculated as surface area / volume | Double |
| max_depth_m | Maximum depth at the deep hole in meters | Double |
| residence_time_yr | Hydrologic residence time of a lake in years. Values are general as residence time varies temporally. | Double |
| known_manipulations | List of known manipulations of the lake enacted by humans. | String |

**Table A2.** LakeBeD-US: Ecology Edition High and Low frequency observation tables metadata

| Column | Description | Data type |
| --- | --- | --- |
| source | Source of a specific data point, in the form [repository, either EDI or NEON] [identifier in repository] | String |
| datetime | Date and time of observation | Timestamp |
| lake_id | Code identifying a lake, matching with the lake information table | String |
| depth | Depth of observation in meters | Double |
| variable | Variable observed. One of `chla` (chlorophyll a), `do` (dissolved oxygen), `fdom` (flourescent dissolved organic matter), `temp` (temperature), `phyco` (phycocyanin), `tp` (total phosphorus), `drp` (dissolved reactive phosphorus), `tn` (total nitrogen), `no2` (nitirite), `no3` (nitrate), `no3no2` (combined nitrite and nitrate), `nh4` (ammonium), `dic` (dissolved inorganic carbon), `doc` (dissolved organic carbon), `poc` (particulate organic carbon), `par` (photosynthetically active radiation), `secchi` (Secchi depth), or `inflow` (discharge rate into lake) | String |
| unit | Unit of observation. One of `RFU` (Relative Fluorescence Units), `MicroGM-PER-L` (micrograms per liter), `MilliGM-PER-L` (milligrams per liter), `DEG_C` (degrees Celcius), `MicroMOL-PER-M2-SEC` (micromoles per meters squared per second), `M` (meters), or `M3-PER-SEC` (cubic meters per second) | String |
| observation | value of observation | Double |
| flag | Quality code of observation. See the flag guide for details. | Integer |



**Table A3.** LakeBeD-US: Ecology Edition Flag Guide table metadata. Citations for source datasets can be found in Appendix B1. Each row of the Flag Guide table corresponds to a definition, where common definitions between flags of data sources are aligned.

| Column | Description | Source Datasets |
|---|---|---|
| LakeBeD-US Flag | Quality flag used in LakeBeD-US | |
| LakeBeD-US Definition | Definition for quality flag used in LakeBeD-US | |
| NTL HF Flag | Quality flag used in high-frequency datasets from NTL-LTER | Magnuson et al. (2023c-h, 2024b-e) |
| NTL HF Definition | Definition for quality flag used in high-frequency datasets from NTL-LTER | |
| NTL LF Flag | Quality flag used in manually sampled datasets from NTL-LTER | Magnuson et al. (2023a-b, 2023i-j, 2024a) |
| NTL LF Definition | Definition for quality flag used in manually sampled datasets from NTL-LTER | |
| VR-LTREB HF Flag | Quality flag used in the high-frequency conductivity, temperature, and depth (CTD) dataset from Virginia Reservoirs LTREB | Carey et al. (2024d) |
| VR-LTREB HF Definition | Definition for quality flag used in the high-frequency conductivity, temperature, and depth (CTD) dataset from Virginia Reservoirs LTREB | |
| VR-LTREB Inflows Flag | Quality flag used in the inflow dataset from Virginia Reservoirs LTREB | Carey et al. (2024b) |
| VR-LTREB Inflows Definition | Definition for quality flag used in the inflow dataset from Virginia Reservoirs LTREB | |
| VR-LTREB Catwalk Flag | Quality flag used in the high-frequency Falling Creek Reservoir water quality dataset from Virginia Reservoirs LTREB | Carey et al. (2024e-f) |
| VR-LTREB Catwalk Definition | Definition for quality flag used in the high-frequency Falling Creek Reservoir water quality dataset from Virginia Reservoirs LTREB | |
| VR-LTREB Water Chem Flag | Quality flag used in water chemistry dataset from Virginia Reservoirs LTREB | Carey et al. (2024g) |
| VR-LTREB Water Chem Definition | Definition for quality flag used in water chemistry dataset from Virginia Reservoirs LTREB | |
| VR-LTREB Secchi Time Flag | Quality flag used for Secchi data from Virginia Reservoirs LTREB | Carey et al. (2024a) |
| VR-LTREB Secchi Time Definition | Definition for quality flag used in Secchi data from Virginia Reservoirs LTREB | |
| VR-LTREB Secchi/YSI Flag | Quality flag used in the sondecast dataset from Virginia Reservoirs LTREB | Carey et al. (2024a) |
| VR-LTREB Secchi/YSI Definition | Definition for quality flag used in sondecast dataset from Virginia Reservoirs LTREB | |
| VR-LTREB Chla Flag | Quality flag used in the filtered chlorophyll a dataset from Virginia Reservoirs LTREB | Carey et al. (2024c) |
| VR-LTREB Chla Definition | Definition for quality flag used in the filtered chlorophyll a dataset from Virginia Reservoirs LTREB | |
| NWT WQ Flag | Quality flag used in water quality datasets from NWT-LTER | McKnight et al. (2021, 2023) |
| NWT WQ Definition | Definition for quality flag used in water quality datasets from NWT-LTER | |
| NWT HF Flag | Quality flag used in high-frequency datasets from NWT-LTER | Johnson et al. (2024a-d) |
| NWT HF Definition | Definition for quality flag used in high-frequency datasets from NWT-LTER | |
| NEON Flag | Quality flag used in NEON datasets | NEON (2024a*-f, 2024h-i) |
| NEON Definition | Definition for quality flag used in NEON datasets | |

*LakeBeD-US lists the maximum depth of a lake for Secchi depth when the Secchi depth hits the bottom. Secchi data from NEON (2024a) lists when the disk hits the lake bottom, and the maximum depth measured, but reports a missing value for Secchi when this happens. In these cases, LakeBeD-US lists the maximum depth as the Secchi depth and applies a flag indicating this substitution was made. This flag does not originate from NEON (2024a).





**Table A4.** Metadata for LakeBeD-US: Computer Science Edition. All possible columns from the high- and low-frequency datasets and 1D and 2D variables are listed.

| Column Name | Description/Water Quality Variable | Units | Dimensionality |
|---|---|---|---|
| datetime | Time of the observation in the lake's local time | | |
| flag | Quality flag for the observed value | | |
| depth | Depth of the observed value | m | |
| chla_rfu | Chlorophyll *a* | RFU | 2D |
| chla_ugl | Chlorophyll *a* | $\mu g\,L^{-1}$ | 2D |
| do | Dissolved oxygen | $mg\,L^{-1}$ | 2D |
| fdom | Flourescent dissolved organic matter | RFU | 2D |
| temp | Temperature | °C | 2D |
| phyco | Phycocyanin | RFU | 2D |
| tp | Total phosphorus | $\mu g\,L^{-1}$ | 2D |
| drp | Dissolved reactive phosphorus | $\mu g\,L^{-1}$ | 2D |
| tn | Total nitrogen | $\mu g\,L^{-1}$ | 2D |
| no2 | Nitrite as nitrogen ($NO_2$-N) | $\mu g\,L^{-1}$ | 2D |
| no3 | Nitrate as nitrogen ($NO_3$-N) | $\mu g\,L^{-1}$ | 2D |
| no3no2 | Combined nitrite and nitrate as nitrogen ($NO_2$+$NO_3$-N) | $\mu g\,L^{-1}$ | 2D |
| nh4 | Ammonium as nitrogen ($NH_4$-N) | $\mu g\,L^{-1}$ | 2D |
| dic | Dissolved inorganic carbon | $mg\,L^{-1}$ | 2D |
| doc | Dissolved organic carbon | $mg\,L^{-1}$ | 2D |
| poc | Particulate organic carbon | $mg\,L^{-1}$ | 2D |
| par | Photosynthetically active radiation (light) | $\mu mol\,m^{-2}\,s^{-1}$ | 2D |
| secchi | Secchi depth | m | 1D |
| inflow | Surface water inflow into the lake | $m\,s^{-1}$ | 1D |



## Appendix B: Citations for data used in LakeBeD-US

### B1 Observational data sources

Carey, C. C., Breef-Pilz, A., Wander, H. L., Delany, A. D., Haynie, G. K., Keverline, R. L., Kricheldorf, M. K., and Tipper, E. M.: Secchi depth data and discrete depth profiles of water temperature, dissolved oxygen, conductivity, specific conductance, photosynthetic active radiation, oxidation-reduction potential, and pH for Beaverdam Reservoir, Carvins Cove Reservoir, Falling Creek Reservoir, Gatewood Reservoir, and Spring Hollow Reservoir in southwestern Virginia, USA 2013-2023 (12), Environmental Data Initiative [data set], https://doi.org/10.6073/pasta/6c27a31ed56662c13016307d0bb99986, 2024a.

Carey, C. C., Breef-Pilz, A., and Delany, A. D.: Discharge time series for the primary inflow tributary entering Falling Creek Reservoir, Vinton, Virginia, USA 2013-2023 (12), Environmental Data Initiative [data set], https://doi.org/10.6073/pasta/510534cd94e9cba40e2b0173e784c2b8, 2024b.

Carey, C. C., Breef-Pilz, A., Hoffman, K. K., Niederlehner, B. R., Haynie, G., Keverline, R., Kricheldorf, M., and Tipper, E.: Filtered chlorophyll a time series for Beaverdam Reservoir, Carvins Cove Reservoir, Claytor Lake, Falling Creek Reservoir, 470 voir, Gatewood Reservoir, Smith Mountain Lake, Spring Hollow Reservoir in southwestern Virginia, and Lake Sunapee in Sunapee, New Hampshire, USA during 2014-2023 (4), Environmental Data Initiative [data set], https://doi.org/10.6073/pasta/bdea148e951b2dd11c74b51854c3aab5, 2024c.

Carey, C. C., Lewis, A. S. L., and Breef-Pilz, A.: Time series of high-frequency profiles of depth, temperature, dissolved oxygen, conductivity, specific conductance, chlorophyll a, turbidity, pH, oxidation-reduction potential, photosynthetically active 475 radiation, colored dissolved organic matter, phycocyanin, phycoerythrin, and descent rate for Beaverdam Reservoir, Carvins Cove Reservoir, Falling Creek Reservoir, Gatewood Reservoir, and Spring Hollow Reservoir in southwestern Virginia, USA 2013-2023 (14), Environmental Data Initiative [data set], https://doi.org/10.6073/pasta/b406e9a104dafb1b91e1ad85a19384db, 2024d.

Carey, C. C., Breef-Pilz, A., Delany, A. D., Hounshell, A. G., Lewis, A. S. L., Wander, H. L., Haynie, G., Kricheldorf, M., and 480 Tipper, E.: Time series of high-frequency sensor data measuring water temperature, dissolved oxygen, conductivity, specific conductance, total dissolved solids, chlorophyll a, phycocyanin, fluorescent dissolved organic matter, turbidity at discrete depths, and water level in Beaverdam Reservoir, Virginia, USA in 2009-2023 (4), Environmental Data Initiative [data set], https://doi.org/10.6073/pasta/31bb6047e0ac367c60a61884338799c4, 2024e.

Carey, C. C., Breef-Pilz, A., Daneshmand, V., Delany, A. D., and Thomas, R. Q.: Time series of high-frequency 485 sensor data measuring water temperature, dissolved oxygen, pressure, conductivity, specific conductance, total dissolved solids, chlorophyll a, phycocyanin, fluorescent dissolved organic matter, and turbidity at discrete depths in Falling Creek Reservoir, Virginia, USA in 2018-2023 (8), Environmental Data Initiative [data set], https://doi.org/10.6073/pasta/7541e8d297850be7c613d116156735a9, 2024f.

Carey, C. C., Howard, D. W., Hoffman, K. K., Wander, H. L., Breef-Pilz, A., Niederlehner, B. R., Haynie, G., Keverline, R., 490 Kricheldorf, M., and Tipper, E.: Water chemistry time series for Beaverdam Reservoir, Carvins Cove Reservoir, Falling Creek



Reservoir, Gatewood Reservoir, and Spring Hollow Reservoir in southwestern Virginia, USA 2013-2023 (12), Environmental Data Initiative [data set], https://doi.org/10.6073/pasta/7d7fdc5081ed5211651f86862e8b2b1e, 2024g.

Hart, J., Dugan, H., Carey, C., Stanley, E., and Hanson, P.: Lake Mendota Carbon and Greenhouse Gas Measurements at North Temperate Lakes LTER 2016 (22), Environmental Data Initiative [data set], https://doi.org/10.6073/pasta/a2b38bc23fb0061e64ae76bbdec656fd, 2022.

Johnson, P., Yevak, S., Dykema, S., and Loria, K.: Chlorophyll-a data for the Green Lake 4 buoy, 2018 - ongoing. (5), Environmental Data Initiative [data set], https://doi.org/10.6073/pasta/2b90eb17f06898359280f68ce140ef47, 2024a.

Johnson, P. T. J., Yevak, S. E., Dykema, S., and Loria, K. A.: Dissolved oxygen data for the Green Lake 4 buoy, 2018 - ongoing. (6), Environmental Data Initiative [data set], https://doi.org/10.6073/pasta/ded48fa1e3851adcd78b744e3d5b49de, 2024b.

Johnson, P., Yevak, S., Dykema, S., and Loria, K.: PAR data for the Green Lake 4 buoy, 2018 - ongoing. (5), Environmental Data Initiative [data set], https://doi.org/10.6073/pasta/cd2a197b4297259428d67c97d32f25b4, 2024c.

Johnson, P., Yevak, S., Dykema, S., and Loria, K.: Temperature data for the Green Lake 4 buoy, 2018 - ongoing. (5), Environmental Data Initiative [data set], https://doi.org/10.6073/pasta/5d1c305fda142f2af462dcdbf77b33ab, 2024d.

Lottig, N.: High Frequency Under-Ice Water Temperature Buoy Data - Crystal Bog, Trout Bog, and Lake Mendota, Wisconsin, USA 2016-2020 (3), Environmental Data Initiative [data set], https://doi.org/10.6073/pasta/ad192ce8fbe8175619d6a41aa2f72294, 2022.

Magnuson, J. J., Carpenter, S. R., and Stanley, E. H.: North Temperate Lakes LTER: Chemical Limnology of Primary Study Lakes: Nutrients, pH and Carbon 1981 - current (60), Environmental Data Initiative [data set], https://doi.org/10.6073/pasta/325232e6e4cd1ce04025fa5674f7b782, 2023a.

Magnuson, J. J., Carpenter, S. R., and Stanley, E. H.: North Temperate Lakes LTER: Chlorophyll - Trout Lake Area 1981 - current (32), Environmental Data Initiative [data set], https://doi.org/10.6073/pasta/4a110bd6534525f96aa90348a1871f86, 2023b.

Magnuson, J. J., Carpenter, S. R., and Stanley, E. H.: North Temperate Lakes LTER: High Frequency Meteorological and Dissolved Oxygen Data - Sparkling Lake Raft 1989 - current (34), Environmental Data Initiative [data set], https://doi.org/10.6073/pasta/9d054e35fb0b8d3a36b49b5e7a35f48f, 2023c.

Magnuson, J. J., Carpenter, S. R., and Stanley, E. H.: North Temperate Lakes LTER: High Frequency Meteorological and Metabolism Data - Crystal Bog Buoy 2005 - present (14), Environmental Data Initiative [data set], https://doi.org/10.6073/pasta/aa8d03b297cc86aaab404e4d25179a1a, 2023d.

Magnuson, J. J., Carpenter, S. R., and Stanley, E. H.: North Temperate Lakes LTER: High Frequency Meteorological and Metabolism Data - Trout Bog Buoy 2003 - present (26), Environmental Data Initiative [data set], https://doi.org/10.6073/pasta/6a281ee14843e7f80fff07e31d6e9cb0, 2023e.

Magnuson, J. J., Carpenter, S. R., and Stanley, E. H.: North Temperate Lakes LTER: High Frequency Water Temperature Data - Crystal Bog Buoy 2005 - current (10), Environmental Data Initiative [data set], https://doi.org/10.6073/pasta/800e42bf5421eb3d601a07245ff5750e, 2023f.



Magnuson, J. J., Carpenter, S. R., and Stanley, E. H.: North Temperate Lakes LTER: High Frequency Water Temperature Data - Sparkling Lake Raft 1989 - current (24), Environmental Data Initiative [data set], https://doi.org/10.6073/pasta/52ceba5984c4497d158093f32b23b76d, 2023g.

Magnuson, J. J., Carpenter, S. R., and Stanley, E. H.: North Temperate Lakes LTER: High Frequency Water Temperature Data - Trout Bog Buoy 2003 - current (29), Environmental Data Initiative [data set], https://doi.org/10.6073/pasta/9535bbc321ebd512cd0e8b0f1d7821be, 2023h.

Magnuson, J. J., Carpenter, S. R., and Stanley, E. H.: North Temperate Lakes LTER: Physical Limnology of Primary Study Lakes 1981 - current (35), Environmental Data Initiative [data set], https://doi.org/10.6073/pasta/be287e7772951024ec98d73fa94eec08, 2023i.

Magnuson, J. J., Carpenter, S. R., and Stanley, E. H.: North Temperate Lakes LTER: Secchi Disk Depth; Other Auxiliary Base Crew Sample Data 1981 - current (32), Environmental Data Initiative [data set], https://doi.org/10.6073/pasta/4c5b055143e8b7a5de695f4514e18142, 2023j.

Magnuson, J., Carpenter, S., and Stanley, E.: North Temperate Lakes LTER: Chlorophyll - Madison Lakes Area 1995 - current (30), Environmental Data Initiative [data set], https://doi.org/10.6073/pasta/da59b1093236ceb67d2cf220b17e5658, 2024a.

Magnuson, J. J., Carpenter, S. R., and Stanley, E. H.: North Temperate Lakes LTER: High Frequency Data: Meteorological, Dissolved Oxygen, Chlorophyll, Phycocyanin - Lake Mendota Buoy 2006 - current (39), Environmental Data Initiative [data set], https://doi.org/10.6073/pasta/daad81be7f12173e3aefbf3df5d6d2fe, 2024b.

Magnuson, J. J., Carpenter, S. R., and Stanley, E. H.: North Temperate Lakes LTER: High Frequency Meteorological and Dissolved Oxygen Data - Trout Lake Buoy 2004 - current (42), Environmental Data Initiative [data set], https://doi.org/10.6073/pasta/1b66ce0b3f0cf7c3f5922fb320b5591e, 2024c.

Magnuson, J. J., Carpenter, S. R., and Stanley, E. H.: North Temperate Lakes LTER: High Frequency Water Temperature Data - Lake Mendota Buoy 2006 - current (33), Environmental Data Initiative [data set], https://doi.org/10.6073/pasta/b6b6b2f2070500202e10e219044b547b, 2024d.

Magnuson, J. J., Carpenter, S. R., and Stanley, E. H.: North Temperate Lakes LTER: High Frequency Water Temperature Data - Trout Lake Buoy 2004 - current (29), Environmental Data Initiative [data set], https://doi.org/10.6073/pasta/767e476fca7bedf46d4905517854c8f7, 2024e.

McKnight, D., Johnson, P., Loria, K., Niwot Ridge LTER, and Dykema, S.: Stream and lake water chemistry data for Green Lakes Valley, 1998 - ongoing. (3), Environmental Data Initiative [data set], https://doi.org/10.6073/pasta/811e22e67aa850fa6c03148ab621e76e, 2021.

McKnight, D. M., Yevak, S., Dykema, S., Loria, K., and Niwot Ridge LTER: Water quality data for Green Lakes Valley, 2000 - ongoing. (9), Environmental Data Initiative [data set], https://doi.org/10.6073/pasta/4835fff2b96f16677ab1abb9c46db34b, 2023.

National Ecological Observatory Network (NEON): Chemical properties of surface water (DP1.20093.001), RELEASE-2024, NEON [data set], https://doi.org/10.48443/FDFD-D514, 2024a.



National Ecological Observatory Network (NEON): Depth profile at specific depths (DP1.20254.001), RELEASE-2024, NEON [data set], https://doi.org/10.48443/VCT8-PR05, 2024b.

National Ecological Observatory Network (NEON): Discharge field collection (DP1.20048.001), RELEASE-2024, NEON [data set], https://doi.org/10.48443/3746-1981, 2024c.

National Ecological Observatory Network (NEON): Nitrate in surface water (DP1.20033.001), RELEASE-2024, NEON 565    [data set], https://doi.org/10.48443/MVDB-K902, 2024d.

National Ecological Observatory Network (NEON): Periphyton, seston, and phytoplankton chemical properties (DP1.20163.001), RELEASE-2024, NEON [data set], https://doi.org/10.48443/25WY-9F31, 2024e.

National Ecological Observatory Network (NEON): Photosynthetically active radiation at water surface (DP1.20042.001), RELEASE-2024, NEON [data set], https://doi.org/10.48443/S71B-KK05, 2024f.

National Ecological Observatory Network (NEON): Photosynthetically active radiation below water surface (DP1.20261.001), RELEASE-2024, NEON [data set], https://doi.org/10.48443/JNWY-XY08, 2024g.

National Ecological Observatory Network (NEON): Secchi depth (DP1.20252.001), RELEASE-2024, NEON [data set], https://doi.org/10.48443/DTR7-N376, 2024h.

National Ecological Observatory Network (NEON): Temperature at specific depth in surface water (DP1.20264.001), 575    RELEASE-2024, NEON [data set], https://doi.org/10.48443/4WDS-5B25, 2024i.

National Ecological Observatory Network (NEON): Water quality (DP1.20288.001), RELEASE-2024, NEON [data set], https://doi.org/10.48443/T7RJ-PK25, 2024j.

**B2   Static lake attributes sources**

Aho, K. S., Maavara, T., Cawley, K. M., and Raymond, P. A.: Inland Waters can Act as Nitrous Oxide Sinks: Observa-580    tion and Modeling Reveal that Nitrous Oxide Undersaturation May Partially Offset Emissions, Geophys. Res. Lett., 50, e2023GL104987, https://doi.org/10.1029/2023GL104987, 2023.

Baron, J. S. and Caine, N.: Temporal coherence of two alpine lake basins of the Colorado Front Range, USA, Freshwater Biol., 43, 463–476, https://doi.org/10.1046/j.1365-2427.2000.00517.x, 2000.

Carey, C. C., Lewis, A. S. L., Howard, D. W., Woelmer, W. M., Gantzer, P. A., Bierlein, K. A., Little, J. C., and WVWA: 585    Bathymetry and watershed area for Falling Creek Reservoir, Beaverdam Reservoir, and Carvins Cove Reservoir (1), Environmental Data Initiative [data set], https://doi.org/10.6073/pasta/352735344150f7e77d2bc18b69a22412, 2022.

Cory, R. M., McKnight, D. M., Chin, Y.-P., Miller, P., and Jaros, C. L.: Chemical characteristics of fulvic acids from Arctic surface waters: Microbial contributions and photochemical transformations, J. Geophys. Res.-Biogeo., 112, https://doi.org/10.1029/2006JG000343, 2007.

Doubek, J. P., Campbell, K. L., Doubek, K. M., Hamre, K. D., Lofton, M. E., McClure, R. P., Ward, N. K., and Carey, C. C.: The effects of hypolimnetic anoxia on the diel vertical migration of freshwater crustacean zooplankton, Ecosphere, 9, e02332, https://doi.org/10.1002/ecs2.2332, 2018.





Flanagan, C. M., McKnight, D. M., Liptzin, D., Williams, M. W., and Miller, M. P.: Response of the Phytoplankton Community in an Alpine Lake to Drought Conditions: Colorado Rocky Mountain Front Range, U.S.A, Arct. Antarct. Alp. Res., 41, 191–203, https://doi.org/10.1657/1938.4246-41.2.191, 2009.

Gaeta, J. W., Hrabik, T. R., Sass, G. G., Roth, B. M., Gilbert, S. J., and Vander Zanden, M. J.: A whole-lake experiment to control invasive rainbow smelt (Actinoperygii, Osmeridae) via overharvest and a food web manipulation, Hydrobiologia, 746, 433–444, https://doi.org/10.1007/s10750-014-1916-3, 2015.

Gerling, A. B., Browne, R. G., Gantzer, P. A., Mobley, M. H., Little, J. C., and Carey, C. C.: First report of the successful operation of a side stream supersaturation hypolimnetic oxygenation system in a eutrophic, shallow reservoir, Water Res., 67, 129–143, https://doi.org/10.1016/j.watres.2014.09.002, 2014.

Lathrop, R. C. and Carpenter, S. R.: Water quality implications from three decades of phosphorus loads and trophic dynamics in the Yahara chain of lakes, Inland Waters, 4, 1–14, https://doi.org/10.5268/IW-4.1.680, 2014.

Lathrop, R. C., Johnson, B. M., Johnson, T. B., Vogelsang, M. T., Carpenter, S. R., Hrabik, T. R., Kitchell, J. F., Magnuson, J. J., Rudstam, L. G., and Stewart, R. S.: Stocking piscivores to improve fishing and water clarity: a synthesis of the Lake Mendota biomanipulation project, Freshwater Biol., 47, 2410–2424, https://doi.org/10.1046/j.1365-2427.2002.01011.x, 2002.

Lawson, Z. J., Vander Zanden, M. J., Smith, C. A., Heald, E., Hrabik, T. R., and Carpenter, S. R.: Experimental mixing of a north-temperate lake: testing the thermal limits of a cold-water invasive fish, Can. J. Fish. Aquat. Sci., 72, 926–937, https://doi.org/10.1139/cjfas-2014-0346, 2015.

Lin, Y.-T. and Wu, C. H.: Response of bottom sediment stability after carp removal in a small lake, Ann. Limnol. - Int. J. Lim., 49, 157–168, https://doi.org/10.1051/limn/2013049, 2013.

Lottig, N. R. and Dugan, H. A.: North Temperate Lakes-LTER Core Research Lakes Information (1), Environmental Data Initiative [data set], https://doi.org/10.6073/pasta/b9080c962f552029ee2b43aec1410328, 2024.

Mrnak, J. T., Sikora, L. W., Zanden, M. J. V., and Sass, G. G.: Applying Panarchy Theory to Aquatic Invasive Species Management: A Case Study on Invasive Rainbow Smelt Osmerus mordax, Rev. Fish. Sci. Aquac., 31, 66–85, https://doi.org/10.1080/23308249.2022.2078951, 2023.

Perales, K. M., Hansen, G. J. A., Hein, C. L., Mrnak, J. T., Roth, B. M., Walsh, J. R., and Vander Zanden, M. J.: Spatial and temporal patterns in native and invasive crayfishes during a 19-year whole-lake invasive crayfish removal experiment, Freshwater Biol., 66, 2105–2117, https://doi.org/10.1111/fwb.13818, 2021.

Rast, W. and Lee, G. F.: Report on Nutrient Load - Eutrophication Response of Lake Wingra, Wisconsin, Environmental Research Laboratory-Corvallis, Office of Research and Development, U.S. Environmental Protection Agency, 1977.

Thomas, R. Q., McClure, R. P., Moore, T. N., Woelmer, W. M., Boettiger, C., Figueiredo, R. J., Hensley, R. T., and Carey, C. C.: Near-term forecasts of NEON lakes reveal gradients of environmental predictability across the US, Front. Ecol. Environ., 21, 220–226, https://doi.org/10.1002/fee.2623, 2023.

The National Map Bulk Point Query Service: https://apps.nationalmap.gov/bulkpqs/, last access: 13 June 2024.



Upadhyay, S., Bierlein, K. A., Little, J. C., Burch, M. D., Elam, K. P., and Brookes, J. D.: Mixing potential of a surface-mounted solar-powered water mixer (SWM) for controlling cyanobacterial blooms, Ecol. Eng., 61, 245–250, https://doi.org/10.1016/j.ecoleng.2013.09.032, 2013.

Watras, C. J. and Hanson, P. C.: Ecohydrology of two northern Wisconsin bogs, Ecohydrology, 16, e2591, https://doi.org/10.1002/eco.2591, 2023.

Webster, K. E., Kratz, T. K., Bowser, C. J., Magnuson, J. J., and Rose, W. J.: The influence of landscape position on lake chemical responses to drought in northern Wisconsin, Limnol. Oceanogr., 41, 977–984, https://doi.org/10.4319/lo.1996.41.5.0977, 1996.

630

*Author contributions.* BJM led data curation, data harmonization for LakeBeD-US: Ecology Edition, manuscript preparation including data visualization, and data product development. AP developed LakeBeD-US: Computer Science Edition, conducted the benchmarking task, and led the authorship of those sections of the manuscript. AP, AN, SF, and AK collaboratively conceptualized the computer science aspects of this manuscript. MEL contributed code to harmonize data from the Virginia Reservoirs LTREB for LakeBeD-US: Ecology Edition. CCC, AK, PCH, and MEL conceptualized the project and selected the included water quality variables. All authors contributed to the revising and editing of this manuscript.

635

*Competing interests.* The contact author has declared that none of the authors has any competing interests.

640

*Disclaimer.* Any use of trade, firm, or product names is for descriptive purposes only and does not imply endorsement by the U.S. Government.

*Financial support.* This project is supported in part by the U.S. National Science Foundation through grants for investigating water quality with knowledge guided machine learning (DEB-2213549, DEB-2213550), the NTL-LTER (DEB-2025982), Virginia Reservoirs LTREB program (DEB-2327030, EF-2318861), and the Environmental Data Initiative (DBI-2223103).

645

*Acknowledgements.* We are grateful to the original authors and curators of the NWT-LTER, NEON, NTL-LTER, and Virginia Reservoirs LTREB datasets for making their data publicly available and upholding the principles of FAIR data. We would like to thank Emily Stanley and Mark Gahler of the NTL-LTER for considering the needs of this project when spearheading the creation of a dataset of lake attributes for the NTL-LTER lakes (Lottig and Dugan, 2024), and for answering questions about NTL-LTER metadata. We would like to thank Aditya Tewari, who assisted with the data wrangling code converting NTL-LTER data to the LakeBeD-US: Ecology Edition format. Monitoring at the Virginia Reservoirs was a team effort uniquely enabled by the Western Virginia Water Authority and we thank all of the researchers

650



past and present who contributed to data collection and publishing (cited in those respective data packages). We would like to thank Hsiao Hsuan Yu and Robert Ladwig, members of our team who contributed ideas and plans of how this dataset could be used for future models and experiments. The authors were financially supported by the U.S. National Science Foundation.



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
