# Peer review of "LakeBeD-US: a benchmark dataset for lake water quality time series and vertical profiles"

_Earth System Science Data, 2025_

## Referee Comment (RC3)

**LakeBeD-US***: a benchmark dataset for lake water quality time series and vertical profiles*

Bennett J. **McAfee** et al.

Correspondence: **Bennett J. McAfee (bennettjmcafee@gmail.com)**

**Abstract.**

Water quality in lakes is an emergent property of complex biotic and abiotic processes that differ across spatial and temporal scales.

**[If the authors mean that the processes vary over spatial and temporal scales then either the number or processes vary or the nature of each process may vary. Then "differ", as the plural form, seems an appropriate choice. However, if the authors mean that the water quality in lakes varies over spatial and temporal scales "differs" as a singular form seems more appropriate.** Can the authors clarify their meaning please?**]**

Water quality is also a determinant of ecosystem services that lakes provide,[no need for a comma after 'provide'] and thus is of great interest to ecologists. Machine learning and other computer science techniques are [insert 'increasingly'] being used to predict water quality dynamics as well as to gain a greater understanding of water quality patterns and controls. To benefit both the sciences of ecology and computer science, we have created a benchmark dataset of lake water quality time series and vertical profiles. LakeBeD-US contains over 500 million unique observations of lake water quality collected by multiple long-term monitoring organizations across 17 water quality variables in 21 lakes in the United States. There are two published versions of LakeBeD-US: an "Ecology Edition" published in the Environmental Data Initiative repository, and a "Computer Science Edition" published in the Hugging Face repository. Each edition is formatted in a manner conducive to inquiries and analyses specific to each domain. For ecologists, LakeBeD-US provides an opportunity to study the spatial and temporal dynamics of several lakes with varying water quality, ecosystem, and landscape characteristics. For computer scientists, LakeBeD-US acts as a benchmark dataset that enables the advancement of machine learning for water quality prediction. [in the previous 2 sentences both refer to the same name 'LakeBeD-US' but each is recognised in the preceding text as either the '"Ecology Edition" or the "Computer Science Edition". To minimise confusion, especially to an "AI" algorithm, there needs to be clarity here and in the whole paper. Would "**LakeBed-US-E**" and **"LakeBeD-US-CS"** suffice as unambiguous acronyms of each program? **Can the authors please clarify and alter the rest of the MS accordingly?**]

**1 Introduction**

[revised manuscript text omitted]

---

## Author Response (AR1)

**Author responses to referee comments for *LakeBeD-US: a benchmark dataset for lake water quality time series and vertical profiles***

**RC1: comment by Dr. Paul S. Blackwell https://doi.org/10.5194/essd-2025-27-RC1**
"The MS is an excellent offering in a complex and demanding area of applied science and i compliment the authors on assembling the doc and even more for the background work associated with the project.
On reading the abstract and Introduction, however there are numerous challenges of interpretation of the US English and needs for clarity and clarification. Additionally the grammar and sentence style varied form section to section and really needs skilled attention to enable uniformity of style, especially to minimise the complexity of some the the more detailed sentences for the 'average' reader of your journal.

The attached doc of my comments is not in MS Word 'review' style, but a copy-paste to Word from the original PDF.

The red sections and alterations are my considered suggestions and deductions.  I hope the content of the doc is sufficiently clear for the authors to understand. If not I will gladly clarify by return email (or whatever method is appropriate in these circumstances)."

Attachment:  📕 Blackwell comments.pdf

**Author response to RC1:**

1. "On reading the abstract and Introduction, however there are numerous challenges of interpretation of the US English and needs for clarity and clarification. Additionally the grammar and sentence style varied form section to section and really needs skilled attention to enable uniformity of style, especially to minimise the complexity of some the the more detailed sentences for the 'average' reader of your journal."
   a. We thank the referee for pointing out the incongruity of style in the manuscript, as it is the case that the co-authors of this manuscript come from multiple academic disciplines, each with their own writing conventions. We have adjusted much of the writing in the computer science-oriented sections to align with a style more typical of environmental science. Most of these changes are in Section 4: the benchmarking task. We have also integrated all of the suggestions present in the attached document regarding sentence structure and abbreviating the dataset names for legibility.

**RC2: comment by Anonymous Referee #2** https://doi.org/10.5194/essd-2025-27-RC2

- Some lake water quality observation datasets in the United States are released in the manuscript. These datasets can be used to study the spatial and temporal dynamics of some lakes with varying water quality, ecosystem and landscape as well as the lake water quality prediction.
- Fig. 1 shows 21 lakes. More information of these lakes should be further provided, like depth, area, elevation, etc.
- In situ observations of these lakes should be preprocessed based on some unified standards. What standards are used? How to process these in situ data?
- How to define the benchmark dataset? What characteristics should be satisfied? How to verify the dataset?
- Some applications to ecosystem, landscape and the lake water quality prediction can be further shown.

**Author response to RC2:**

1. "Fig. 1 shows 21 lakes. More information of these lakes should be further provided, like depth, area, elevation, etc."
   a. We have added a new table, Table 1 in the revised manuscript, containing the attributes requested and more. Fig. 1 was revised to not include a table to prevent repeating the same information that appears in Table 1.
2. In situ observations of these lakes should be preprocessed based on some unified standards. What standards are used? How to process these in situ data?
   a. The standards used for the processing of data are described in Section 2.1.4, wherein the data quality flags are harmonized from across various data sources. This process is designed to allow flexibility for users of the dataset to filter the dataset to their preferred level of uncertainty. We offer a suggested set of acceptable quality flags for benchmarking tasks in Table 6. To emphasize this point, we have added a reference to Table 6 within Section 2.1.4.
3. How to define the benchmark dataset? What characteristics should be satisfied? How to verify the dataset?
   a. There is no definitive list of criteria for what makes a benchmark dataset. However, we have followed the examples set by Sarkar et al. (2020, *Database*) and Weinstein et al. (2021, *PLOS Computational Biology*). We originally cited these works, but have revised the introduction to include a summary of the main criteria and adapted their definitions to be relevant to ecology and lake water quality prediction.
4. Some applications to ecosystem, landscape and the lake water quality prediction can be further shown.
   a. Thank you for pointing this out. We have added "Section 5.2: LakeBeD-US for ecological applications" to the Discussion section of the manuscript. This section discusses many potential ways LakeBeD-US could be used that would advance scientific understanding of lake water quality dynamics.

**RC4: Comment by Anonymous Referee #3 https://doi.org/10.5194/essd-2025-27-RC4**

Review comments

This paper describes a lake water quality baseline dataset called LakeBeD-US, which integrates long-term monitoring data from 21 lakes in the United States to provide a baseline for lake water quality time series and vertical profile studies. This paper describes in detail the construction process of data set, data structure and application cases in machine learning. The construction of this dataset is of great significance for advancing the study of lake water quality, providing a valuable data resource for ecologists and computer scientists. The article is rich in content, clear in logic, rigorous in method, and has high academic value.

1.    Advantages

Innovative and valuable dataset: The LakeBeD-US dataset is the first benchmark dataset dedicated to time series and vertical profiles of lake water quality, filling a gap in the field. It contains more than 500 million unique water quality observation data, covering 17 water quality variables, with extremely high data volume and diversity, providing rich data support for relevant research.

Multidisciplinary integration: The dataset is not only suitable for ecological research, but also optimized for computer science, which can promote interdisciplinary cooperation between ecology and computer science, and promote the application of knowledge-guided machine learning (KGML) in lake water quality research.

Data integrity and standardization: The author describes in detail the source of the data, the processing process and the quality control method to ensure the accuracy and reliability of the data. At the same time, the data set follows the FAIR principle, which is easy to find, access, interoperate and reuse.

Practicability of application cases: Taking Lake Mendota as an example, this paper demonstrates how to use LakeBeD-US: Computer Science Edition to make multivariate time series prediction, verifies the feasibility and effectiveness of data set in machine learning tasks, and provides reference for subsequent research.

1.    II. Disadvantages

Data representation issues: Although the dataset covers 21 lakes, 13 of them are located in Wisconsin, which may limit the geographic representation of the data to some extent. It is suggested that the authors consider including more lake data from different geographic regions in subsequent work to improve the general applicability of the dataset.

Processing of missing data values: It is mentioned that there are a large number of missing values in the data set, and although this is a common problem in environmental data, the authors adopt the SAITS method in processing missing values, and do not discuss its impact on the results in detail. It is suggested that the authors further analyze the potential impact of missing values on model performance and explore other possible treatment methods.

Adequacy of model validation: In the machine learning application case, the authors used the LSTM-RNN model for prediction, but only reported normalized and unstandardized RMSE metrics. It is recommended to add more model validation indicators, such as coefficient of determination ($R^2$), mean absolute error (MAE), etc., to evaluate the performance of the model more comprehensively.

Generalizability of the conclusions: The conclusions of this paper are mainly based on data from Lake Mendota and may not be fully generalizable to other lakes. It is suggested that the authors further explore the applicability and limitations of the data set and model across different lakes in the discussion.

Data update and maintenance: The timeliness and accuracy of data sets depend on long-term updates and maintenance. It is recommended that the author clarify the plan and mechanism of data update to ensure the continuous availability of the data set.

III. Comprehensive Suggestions

The LakeBeD-US data set proposed in this paper has important scientific value and application prospect, and it is suggested that the author should modify and perfect it according to the above opinions, so as to further improve the quality and influence of the paper. In particular, it is necessary to strengthen the discussion of data representation, model validation and generalization of conclusions, so as to provide readers with more comprehensive and reliable research results.

The following is a more professional review of this paper, which aims to help the research be more scientific and rational:

1.    Data quality control

Data cleaning details: It is mentioned that the data set is cleaned and the format is converted, but the specific data cleaning method is not detailed. It is recommended that the author describe the data cleaning process in detail, including how to deal with missing values, outliers, duplicate values, etc., and what algorithms or tools are used to carry out data cleaning to ensure the accuracy and reliability of the data.

Data quality assessment: Authors should provide more detailed data quality assessment indicators and methods, such as data integrity, accuracy, consistency, etc. A data quality assessment tool or framework can be adopted to systematically assess data quality and report the assessment results in a paper.

Data traceability: In order to ensure the traceability of the data, it is recommended that the author record the metadata information such as the source, collection time, collection method, and processing step of the data in detail, and publish it together with the data set, so that other researchers can clearly understand the generation process and quality status of the data.

1.    Data set construction method

Data fusion method: The paper mentions the fusion of data from different sources, but does not specify the specific method and algorithm of data fusion. It is recommended that the author describe the process of data fusion in detail, including how to resolve conflicts between different data sources, how to align and integrate data, etc., to ensure the consistency and availability of data sets.

Data sampling strategy: The author should further explain the strategy and method of data sampling, including sampling frequency, time range, spatial coverage, etc. In particular, for the sampling of high and low frequency data, it should be explained how to ensure that the sampling adequately captures the dynamic changes in lake water quality while avoiding data redundancy.

The science of data set construction: It is suggested that the author discuss the science and rationality of data set construction in more depth, such as whether the representation of lakes, randomness and systematization of data are considered. Statistical analysis methods or machine learning techniques can be employed to verify that the construction of data sets meets scientific requirements.

1.    Model verification and evaluation

Model selection basis: In the machine learning application case, the author chose the LSTM-RNN model for prediction, but did not specify the basis and advantages of choosing this model. It is recommended that the authors analyze and compare the applicability of different models in the prediction of lake water quality in more depth, explain why the LSTM-RNN model is chosen, and discuss its advantages and disadvantages compared with other models.

Model validation indicators: Currently, only standardized and unstandardized RMSE are used as model validation indicators. It is suggested to add more validation indicators, such as coefficient of determination ($R^2$), mean absolute error (MAE), root mean percentage error (RMSPE), etc., in order to evaluate the performance and generalization ability of the model more comprehensively.

Interpretability of the model: To improve the interpretability of the model, authors are advised to use methods such as feature importance analysis, locally interpretable model-independent exPlanations (LIME), or SHapley Additive exPlanations (SHAP) to explain the model's predictions and key influencing factors.

1.    Generalizability of conclusions

Multi-lake validation: The current conclusions are mainly based on the data of Lake Mendota. It is suggested that the author validate and apply the model in more lakes to test the universality of the conclusions and the generality of the model. This can be done in collaboration with other lake research institutions or by leveraging existing lake datasets.

Applicability under different environmental conditions: Considering the differences of lakes in different geographical regions, climatic conditions, ecological types, etc., it is suggested that the author explore the applicability and performance of the model under different environmental conditions, and analyze the limitations and improvement directions of the model.

Long-term forecasting ability: The current forecasting task mainly focuses on short-term forecasting, and it is suggested that the authors further study the performance of the model in long-term forecasting and how to improve the long-term forecasting ability of the model, which is of great significance for lake water quality management and policy making.

The following are the review comments on the language spelling errors in this paper:

1."LakeBeD-US contains over 500 million unique observations of lake water quality collected by multiple long-term  monitoring organizations across 17 water quality variables in 21 lakes in the United States. "It is suggested to replace" in "with" across ", To more accurately represent data coverage.

2."Water quality varies across spatial and temporal scales due to a variety of interacting physical and biological processes. "It is suggested that" due to "be changed to" as a result of "to enhance the formality of the sentence.

3."For example, hypolimnetic anoxia (low oxygen) in lakes decreases habitat for cold-water fish species. "It is suggested to change" decreases "to" reduces ", To express the meaning of reduction more accurately.

4."Water quality is also a determinant of ecosystem services that lakes provide,  and thus is of great interest to ecologists. "It is suggested to change" is of great interest "to" is highly significant "to enhance the strength of the expression.

5."To meet this challenge, we need scalable water quality models that are supported by observational data of sufficient spatiotemporal resolution  to recreate key water quality dynamics. "It is suggested that" recreate "be changed to" reproduce "to more accurately express the meaning of the reproduction.

Author Response:

Thank you for your very thorough analysis of our paper, we greatly appreciate your time and expertise. Many of your suggestions would benefit the paper, and we addressed as many as we could. However, given the delay in receiving this review and the tight deadline for submission of the finalized manuscript, we cannot feasibly integrate all of the suggestions, even if we were given the possible two-week extension by the editor. But, we would not have submitted this paper if we were not confident that our manuscript contains everything

a reader would need to make effective use of LakeBeD-US. Below are our responses to specific points regarding what is already present in the paper, what we have added in response to your suggestions, or what suggestions are outside of this paper's scope.

- Regarding data representation issues, processing of missing data values, adequacy of model validation, and generalizability of the conclusions
  - We thank the referee for making a very thorough list of potential improvements to the dataset and the benchmarking task presented in this paper. This dataset can certainly be improved over time in various ways. Data representation can be addressed by the introduction of additional modules proposed in Section 5.4 of the revised manuscript, which outlines our plans for dataset maintenance. Additional model validation and generalizability across the dataset is addressed in future works of our group, and is outside the scope of this paper.
- Regarding data update and maintenance
  - The enduring availability of this dataset is ensured by the repositories in which they are stored. Even if the duties of the corresponding author and the NTL-LTER information manager (as listed in the metadata of LakeBeD-US: Ecology Edition) are unable to be fulfilled in the maintenance of the dataset, the source code to update the dataset in an automated fashion is published alongside the data itself. This allows users to update the dataset for themselves with relatively low effort if needed.
- Regarding data cleaning details
  - Flagging systems from LakeBeD-US: Ecology Edition are carried over from the data sources, each of which has their own data cleaning procedure. The cleaning conducted for LakeBeD-US: Computer Science Edition is described in Section 2.2.1.
- Regarding data quality assessment
  - A data quality assessment is a strong suggestion, however this process would be repetitive of the quality control and data flagging measures that are part of the protocols in the source data packages. We also wish to respect the user's discretion in defining what 'data quality' is. However, deduplication and consistency in categorical variable codes is ensured by the automated data harmonization procedure.
- Regarding data traceability
  - All provenance metadata, and the source code to harmonize the data, is provided alongside the dataset as described in Table 2 and Section 2.1.3 of the revised manuscript.
- Regarding Data fusion method
  - The manual reformatting of each data source is described in Section 2.1.1.
- Regarding Data sampling strategy
  - Tables 3, 4, and 5 in the revised manuscript explain the sampling timeline and strategy for the lakes in LakeBeD-US. Understanding the required temporal

frequency to address specific ecological phenomena is a task that is beyond the scope of this paper, as the possible phenomena that could be investigated are endless. We provide the maximum temporal frequency available for a given lake to empower users to make that decision for their study. Users of the data can use tools like rarefaction analysis to assess the required frequency to address phenomena of interest.

- Regarding the science of data set construction
  - The representativeness and science-informed design of LakeBeD-US is discussed in the introductory paragraphs of Section 2 as well as Section 5.1. Establishing a definition of "scientific requirements" for an ecological dataset is a task that is outside the scope of this paper.
- Regarding model verification and evaluation
  - An LSTM-RNN was used and analyzed with RMSE as these are commonly used in time series prediction and provide an excellent baseline for a benchmark. Comparisons to other models, as well as the use of explainable AI tools to interpret a model, are outside of the scope of a data description paper. Future work from our group addresses these, as multi-model comparison and model interpretability methods are topics deserving their own dedicated works.
- Regarding generalizability of conclusions
  - Similar to above, analysis of the generalizability of the model is a topic complex enough to warrant its own paper that presents novel scientific findings, rather than one intended to introduce a novel dataset.
- Regarding language spelling errors
  - Thank you for these suggestions. We have implemented suggestions 2, 3, and 5. We modified Suggestion 1 slightly to avoid the repetition of the word 'across' within the sentence, instead using 'from' to capture the intent of the suggestion. Suggestion 4 was not implemented to maintain the accuracy of the original statement and keep a consistent tone with the rest of the manuscript.